# Diurnal, seasonal and annual variations of fair weather atmospheric potential gradient, and effects of reduced number concentration of condensation nuclei on PG and air conductivity from long term atmospheric electricity measurements at Świder, Poland

Izabela Pawlak[1], Anna Odzimek[1], Daniel Kępski[1], and José Tacza[1]

[1]Institute of Geophysics, Polish Academy of Sciences, Księcia Janusza 64, 01-452 Warsaw, Poland

**Correspondence:** Izabela Pawlak (izap@igf.edu.pl)

**Abstract.** The atmospheric potential gradient (PG) has been measured at ground level with a radioactive collector method in Stanisław Kalinowski Geophysical Observatory in Świder (52.12°N, 21.23°E), Poland, for several decades. The long term Świder measurements analysed previously revealed rather typical behaviour in the diurnal and seasonal variations of the PG of a land station, assumed to be controlled by pollution. Observations of the PG at such a station usually show a maximum at local winter months which are affected by anthropogenic pollution the most. Recently digitised series of 1965–2005 hourly data has been newly analysed to describe the Świder PG variations in greater detail, also in connection with an analysis of simultaneous measurements of condensation nuclei (CN) measured at 6, 12, 18 UT. An attempt is made to calculate the diurnal and seasonal variations at the CN number concentrations below 10000 cm$^{-3}$. There is a decrease of the PG in the diurnal variation by up to 11% in the winter, and no significant change in the summer. The reduction in the annual variation is 11–26% with the biggest difference in February. In the summer months this difference is negligible. Despite the efforts to minimise the aerosol effect on the PG in the measured CN range, the character of the seasonal and annual variation of PG preserves its character with a maximum in the Northern Hemisphere winter and the minimum in the summer as observed at other mid-latitude stations in this part of the globe. When investigating the effect of reduced CN concentrations on the measured positive conductivity (PC), an increase of 7–17 % is found. Except an additional mechanism affecting the PG in the summer there may be another aerosol fraction outside of range of the condensation nuclei like dust which affects the conductivity, and indirectly the annual variation of the PG.

## 1 Introduction

### 1.1 Diurnal and seasonal variation of GEC as deduced from the atmospheric electricity stations observations

The Global Atmospheric Electric Circuit (GEC) manifests itself in the atmospheric potential gradient (PG) observed at any location on the globe (Rycroft et al., 2000). Diurnal and seasonal variations of the PG are the earliest recognised changes of the PG (e.g., Witkowski, 1902; Israël, 1973b). It has been also early established that even in fair weather (FW) conditions the curves of the diurnal variation had different shapes depending on the location of stations, season and local conditions (Israël,

1973a). At land stations a double oscillation was often observed in the PG variation, present in at least one (usually the warm) season. For example, at Kew, Uppsala and Tokyo, the diurnal variation of the PG exhibits the double oscillations at any time of the year while at Helsinki, Tortosa or Vassijaure it had the double oscillation only in the local summer. At polar stations and over the oceans a single oscillation is usually observed throughout the year and its character is unitary in the Universal Time which has been firstly recognised by Hoffmann (1924). The average fair weather variation of the PG calculated from 1920–1925 observations over the oceans on board of the Carnegie Institution of Washington research vessel "Carnegie" is known as the Carnegie curve (Parkinson and Torreson, 1931; Israël, 1973a; Harrison, 2013). This curve is considered to reflect the variation of the diurnal activity of the global circuit and is often compared with the diurnal variation of the PG observed in fair weather conditions in other places on the globe. In the new century such analyses have been reported by (e.g., Kubicki et al., 2016; Burns et al., 2017; Nicoll et al., 2019; Tacza et al., 2020; Michnowski et al., 2021, and others). In the seasonal or annual variation many observations indicated a winter maximum in the atmospheric potential gradient compared to the summer (e.g. Israël, 1973a; Mani and Huddar, 1972).

Adlerman and Williams (1996) investigated the connection between the PG local winter maximum and more intensive anthropogenic emission of Aitken nuclei which occurred independent of the station's location in Northern or Southern Hemispheres. The annual variation of the air conductivity affected by the attachment of ions to aerosol particles which minimises during the local winter as compared to the local summer supported this relationship. A question arises then when the true maximum of the GEC is (Williams, 2009), and Adlerman and Williams (1996) argued this should be investigated based on observations in locations relatively free from aerosol particles effects. Their analysis of Mauna Loa air-Earth current data and reanalysis of Carnegie and Maud cruises PG data indicated a GEC maximum in the Northern hemisphere summer in agreement with the maximum of global thunderstorm activity (e.g. Christian et al., 2003).

In this paper we reanalyse the diurnal, seasonal and annual variation in the atmospheric potential gradient at the land station Świder (Poland) which also has condensation nuclei monitoring. Previous analyses of the diurnal variation of the PG indicate that Świder is characterised by the double oscillation prevalent in the summer (Northern Hemisphere summer), and the single oscillation in the winter (Kalinowski, 1932). Furthermore, in the annual variation, the PG values in the winter are higher in amplitude (fluctuation level of 30% of the mean), while the air conductivity variability is higher in the summer (Kubicki et al., 2007). This could be explained by the higher emissions of dust and aerosol affecting the local electric conductivity in the winter (Kubicki et al., 2003, 2007), with the dust having the largest fluctuation over the year of 60% of the mean, and aerosols only 20% of the their mean. The thermal convection as well as the dependence of dust concentrations on the meteorological conditions is indicated as a significant factor affecting the variations of the PG and conductivity.

Here we particularly analyse these variations at low levels of the nuclei number concentration which are possible at Świder, in order to investigate whether, with such a limit applied, this may have any considerable indirect effect on the PG by changes in the air conductivity dependent on the concentration of the nuclei. We consider concentrations of the order of several thousand particles, or below 10000 particles per $cm^{-3}$, which are characteristic for rural, non-urban and remote regions of the globe (e.g. Landsberg, 1938; Mohnen and Hidy, 2010). For example, Schonland (1953, Table II) cites the results of CN measurements from Heligoland (6300 $cm^{-3}$) in the conditions of wind from the direction of land. Junge (1951) presents the results of Landsberg

(1938) in a different way, with 9500 cm$^{-3}$ being an average CN concentration for an open country, and 5900 cm$^{-3}$ a mean minimum concentration for a town — these are the conditions we aim to investigate. It is also worth noting the remark by Podzimek et al. (1982) that experiments showed that up to a value of 10000 particles in cm$^3$, the readings of different particle counters were highly consistent and the chemical composition of the aerosol became less important for their readings. This further facilitates comparison of the results obtained in this study with those of other studies conducted under low particle concentration conditions, despite different instruments used for the particle count measurements. It is important to note that our maximum CN threshold is of one order higher than levels found in the cleanest regions of the globe like the Arctic or the Antarctic where CN concentrations are often of the order of several hundred in one cubic centimetre, although higher concentrations also occur there (Kubicki et al., 2016; Karasiński and Kubicki, 2024). Therefore, it is expected that we cannot remove completely the influence of anthropogenic pollution in the PG values.

## 2 Świder published atmospheric electricity data sets

Stanisław Kalinowski Geophysical Observatory of the Institute of Geophysics, Polish Academy of Sciences was founded as the Magnetic Observatory, and since 1929, renamed Geophysical Observatory, also worked as an atmospheric electricity station except during the World War II till 1947. Most of the pre-war electric measurements were lost, and publication of new results of the resumed electric measurements started in 1958. Measurements reports from years 1957–2005 were subsequently published on paper in the form of yearbooks with tabulated data. The first and last paper issue were Warzecha (1960) and Kubicki (2006). The list of all yearbooks providing data for this study can be found in Appendix A. Since the beginning of the atmospheric electric measurements at Świder the PG and conductivity were the main variables to be measured, however only PG measurements were started initially in 1929. After 1949 the observations were complemented by measurements of the air conductivity. Initially these measurements were conducted three times a day (see Sec. 2.3); both polarities were measured and reported in the yearbooks until 1964. In 1965 measurements of the air positive conductivity (PC) were given a priority and its hourly values have been reported in the yearbooks continuously since then (Warzecha, 1968), although the negative conductivity measurements have not been withdrawn and some unpublished regular paper records from the period still exist in the archives of the observatory (M. Kubicki, personal communication, 2024). In this paper we aimed at analysing of the PG and PC, and some effects CN have on them, with emphasis on PG since it is the most commonly measured parameter of atmospheric electricity worldwide. In the final sections we also consider shorter datasets for which both positive and negative conductivity records are available.

### 2.1 Site location

Świder is located in the central part of Poland in the Warsaw suburban area, about 25 km south-east of Warsaw (52.12°N, 21.23°E). It used to be a popular holiday and health resort village located on the Świder river. The distance to the nearest urban center, which is the district town of Otwock, is 2.5 km. There is no major industry in the area but there are local anthropogenic sources of air pollution from household heating, very typical for these suburban conditions. The architecture is dominated by

residential buildings and mainly includes single and multifamily houses. The Observatory is located in a less populated area nearer the river. It includes the main office and three observation pavilions, and two residential buildings in a distance. The entire area covers about 7 ha and is partly covered and surrounded by, predominantly, pine trees, with several clearings. In one of these clearings of an area of approximately 1 ha, one pavilion and the station's instruments for atmospheric electricity and meteorology observations are located.

## 2.2 Potential gradient and conductivity data

The measurements of fair weather atmospheric potential gradient at the Świder observatory have been made with radioactive collectors (the activity of about $30 \, \mu Ci$). Initially two independent collectors were used, each consisting of the radioactive sonde at a height of about 2 m connected to an electrometer and a recording milliammeter. At the beginning Benndorf electrometers were used, and in 1964 replaced by observatory-built electrometers (Warzecha, 1968, 1976). After 2000 they were replaced by commercially available electrometers. The recording milliammeter was also later replaced by an A/D logger. The PG measurements have absolute calibration and reduction to free plane values (e.g., Kubicki, 2006, and earlier yearbooks). The PG was reported as the atmospheric electric field strength rather, in agreement with the atmospheric electricity convention of its sign (positive value indicates downward direction of the vector). The air conductivities due to small ions have been measured using Gerdien tube apparatus, in 1965 replaced with observatory-built aspiration condensers, at 1.4 m height. Hourly averages for each month of the PG measured with the collector as well as the positive conductivity, PC, were calculated and the tabulated data published in the yearbooks.

## 2.3 Condensation nuclei data

After continuous atmospheric electricity observations were resumed after the war also simultaneous regular measurements of aerosol particle number concentrations of the condensation nuclei type (CN, particle size from 0.005 $\mu$m to 10 $\mu$m) have been started (Warzecha, 1960, 1974).

These measurements were carried out using two types of counters: initially using a small Scholz counter, and from 1982 with a photoelectric CN counter built in the observatory which used a chamber of a Verzár counter as a base (the Verzár counter was previously used for occasional continuous observations). The Scholz and Verzár counters are described in more detail by Grabovskiy (1956). The measurement method used by these counters is the process of condensation of water vapor on atmospheric aerosol particles present in the measurement chamber, followed by a quantitative analysis of the resulting mist droplets. A Scholz counter is a type of condensation counter constructed by Scholz as an improvement of the Aitken counter, designed to measure the total concentration of condensation nuclei or nearly the total concentration of aerosol. The main part of the small Scholz counter was a brass cylindrical chamber with a volume of 102 cm$^3$ and a height of 4 cm; the adiabatic volume expansion ratio was 1.25 (rather than the usual standard of 1.2 (Grabovskiy, 1956)). According to Miller and Bodhaine (1982), at such an expansion ratio the supersaturation is in the range 2.93–3.40 for initial temperatures between 25°C and 10°C, respectively, which is equivalent to the supersaturation between 3.93–4.40 according to Wilson (1897). The Scholz counter allows measurements in a wide range of CN concentrations from 5 to 960 000 particles in cm$^3$, and its experimental

error should not be higher than the experimental error of the Aitken counter which is about 10% (Grabovskiy, 1956; McMurry, 2000). Measurements with the Scholz counter were repeated five times, and one measurement could take several minutes. At Świder they were performed using the suction method of an air sample of 1 cm$^3$ in volume taken within the clearing near a meteorological station.

Since January 1983, the measurements have been performed with the photoelectric CN counter that was placed inside a measurement pavilion. The basic measurement of the number of CN took place in a cylindrical chamber filled with the tested air sample of the volume equal to 680 cm$^3$. Estimates of the number of droplets were obtained using a photoelectric counter system by measuring the extinction of light. The air samples were collected from the outside of the building at a height of 1 m above the ground. The suction of air was made through a 1 m long rubber pipe using an electric rotational pump. The measuring range of the counter was 4500 to 850 000 CN in 1 cm$^3$. The electronic circuit system was built (also patented) by Stanisław Warzecha. The measurement accuracy was 15% (Warzecha and Warzechowa, 1979). The yearbooks or other published reports known do not provide any details on the cross check of measurements during the instrument transition period.

The measurements were made at a height of 1 m, except in 1998–1999 at 3 m. Observations were performed three times a day: first between 6:10 UT and 6:30 UT (5:50 – 6:20 UT till 1971), next between 11:00 UT and 11:30 UT, and the last between 18:10 UT and 18:30 UT (19:00 – 19:30 UT till 1971). The average results of the measurements at the three terms referred to in the yearbooks as 6, 12, and 18 UT, respectively, have been included in the observatory yearbooks.

## 2.4 Criteria of fair weather conditions

In order to investigate global signals of atmospheric electricity and minimise the effects of local weather, criteria for fair weather conditions are used (e.g. Imyanitov and Chubarina, 1967). Since 1965 a WMO standard for the fair weather conditions has been applied at Świder: low cloudiness up to 3/8 in okta scale, no precipitation or fog, wind speed less than 6 m/s and PG amplitude less than 1000 V/m, and no negative PG values (e.g., Warzecha, 1968; Kubicki, 2006). The criteria for fair weather conditions were assessed by the observatory staff on an ongoing basis. Foul and fair weather hours are marked in the yearbooks' PG or PC monthly tables.

## 3 Overview of fair weather PG, PC and CN data

In our analysis we consider all fair weather hourly values of the potential gradient (PG) and the condensation nuclei concentrations (CN) from 6, 12, 18 UT published in the Observatory yearbooks 1965–2005 (no criterion regarding the minimum number of points in a month has been applied). There is a total of 131634 hourly PG values with a subset of 16072 hourly values with corresponding CN concentrations and 3303 missing data (2.5%). Table B1 and Table B2 in the Appendix B include the number of FW hourly average mean values of PG and number of FW days (24 h) for each month and year of the analysed period. There are more FW values during the local warmer seasons from March to September. The percentage of FW hours is about 37% on average (see also Table 1) and varies between 49% in the summer and 23% during winter. The total number of FW

**Table 1.** Number of total hours for the period January 1965 – December 2005, number of all fair weather (FW) hours, FW hours with a PG value, and number of the hours or PG values corresponding to all CN observations, and to the CN concentrations below 10000 cm$^{-3}$, 8000 cm$^{-3}$, 6000 cm$^{-3}$ and 4000 cm$^{-3}$, respectively. The percentages in columns refer to the number in bold in the first raw of the table according to the position of the percentage in the column (or to the number in bold in the same colour).

| All hours 1965-2005 | All FW hours 1965-2005 | FW PG hours 1965-2005 | FW PG all CN | FW PG CN<10000 | FW PG CN<8000 | FW PG CN<6000 | FW PG CN<4000 |
|---|---|---|---|---|---|---|---|
| **359400** | **134937** | **131640** | **16074** | **3593** | **2025** | **915** | **122** |
| | 37.54% | 36.62% | 4.47% | 1.00% | 0.56% | 0.25% | 0.03% |
| | | 97.50% | 11.91% | 2.66% | 1.50% | 0.69% | 0.09% |
| | | | 12.21% | 2.72% | 1.54% | 0.71% | 0.09% |
| | | | | 22.35% | 12.60% | 5.69% | 0.76% |

days is 1414. The monthly percentage of FW days is about 9% on average and rises to 14% during summer and decreases to 4% during winter.

The time variations of monthly mean values of PG and CN during this period is shown in Fig. 1. There seems to be several periods of monotonic increase and decrease of the PG, and a general trend of increase of the PG by 2.5 V/m per year. The general trend for CN concentration is a decrease by –120±20 cm$^{-3}$ per year. Analysis of the long-term variation in these series will be a subject of another study.

Histogram of these values and main statistical parameters of the set are given in Fig. 2 and in the last column of Fig. 2. The mean average, Ma, of the total fair weather PG of 1965–2005 is 231 V/m, and the standard deviation, SD, is 131 V/m. The median, Me, is 207 V/m, the first quartile, $Q_1$, equals 139 V/m, and the third quartile, $Q_3$, 294 V/m. The kurtosis, K, equals 5.4 and the skewness, S, is 1.3 what characterises a leptokurtic and right-skewed distribution. These parameters describe the total FW PG set but in the next sections we concentrate on the subset of PG hours for which simultaneously measured CN data also exist, and particularly on the subset for limited CN concentrations less than 10000 cm$^{-3}$. The statistical parameters of the distribution of these subsets are also given in Fig. 2. The selection of PG data measured at the same time as CN creates a new distribution with slightly higher mean (249 V/m) or median value (224 V/m) but similar kurtosis and skewness of ∼5.5 and ∼1.2, respectively. When we further limit the CN concentrations to less than than 10000 cm$^{-3}$ the mean and median value are lower, 221 V/m and 208 V/m, respectively. It also affects the kurtosis which in this case equals 4.7 and skewness is 1.0. Selection of PG-CN pairs limits the number of values, $N$, to 16074 (12.21%) and with the condition of CN lower than 10000 cm$^{-3}$ the number of pairs decreases to 3593 (2.72%) as indicated in Tab. 1.

In Fig. 3 we show a histogram of CN concentrations in fair weather conditions. The mean value of all FW CN concentrations is 18970 cm$^{-3}$ and the standard deviation is 13270 cm$^{-3}$ but there is almost twice more counts in the summer compared to winter. The median is 15400 cm$^{-3}$, the first quartile is 10290 cm$^{-3}$ and the third quartile is 22650 cm$^{-3}$. The kurtosis is equal to 8.8 and the skewness 2.1 what indicates a leptokurtic and right-skewed distribution. The total number of the FW values

**Table 2.** Statistical measures of the distribution of all values FW PG and CN in the series 1965–2005.

| | Winter | | Spring | | Summer | | Autumn | | ALL CN | | ALL FW |
|---|---|---|---|---|---|---|---|---|---|---|---|
| | PG | CN | PG | CN | PG | CN | PG | CN | PG | CN | PG |
| | (V/m) | ($cm^{-3}$) | (V/m) | ($cm^{-3}$) | (V/m) | ($cm^{-3}$) | (V/m) | ($cm^{-3}$) | (V/m) | ($cm^{-3}$) | (V/m) |
| Ma | 351 | 20930 | 250 | 21180 | 193 | 15330 | 253 | 20020 | 249 | 18970 | 231 |
| Sd | 165 | 12800 | 129 | 14940 | 84 | 11220 | 130 | 12970 | 134 | 13270 | 130 |
| Me | 336 | 18200 | 228 | 16900 | 185 | 12560 | 235 | 16740 | 224 | 15400 | 207 |
| $Q_1$ | 235 | 11900 | 162 | 11330 | 131 | 8800 | 160 | 11330 | 155 | 10290 | 139 |
| $Q_3$ | 448 | 26000 | 315 | 26000 | 242 | 18200 | 328 | 24200 | 315 | 22650 | 294 |
| S | 0.6 | 1.7 | 1.2 | 1.9 | 0.8 | 2.9 | 1.0 | 1.9 | 1.2 | 2.1 | 1.3 |
| K | 3.2 | 7.4 | 5.7 | 7.0 | 4.8 | 15.2 | 4.7 | 7.9 | 5.3 | 8.8 | 5.4 |

is 16756 and the subset below 10000 $cm^{-3}$ (see black frame in Fig. 3) counts 3736 which corresponds to 22% of the total. Table 2 and Fig. 4 show the main statistical measures of the FW PG and CN distributions discussed above and separately for each season. The black line separates the distributions of CN concentrations below 10000 $cm^{-3}$.

The lowest concentrations of CN in fair weather conditions (the minimum is 1400 $cm^{-3}$) are observed in the summer with the mean value of ~15000 $cm^{-3}$. In the other seasons the mean value is close to 20000 $cm^{-3}$ with the highest standard deviation values in the spring ~15000 $cm^{-3}$ and the lowest ~11000 $cm^{-3}$ in the summer, and ~13000 $cm^{-3}$ in the winter and in the autumn. In terms of kurtosis and skewness the values in the spring, autumn and winter are similar and vary between 7.0-7.9 and 1.7-1.9, respectively. In the spring these parameters are higher reaching values 15.2 and 2.9.

## 4  Diurnal and seasonal variation of the potential gradient in Świder, 1965-2005

In this section we analyse the diurnal and seasonal variation of Świder FW PG and how it relates to different levels of CN concentrations. Previous works on the analysis of diurnal and seasonal variations of PG focused mainly on full 24 hours of FW, i.e. an FW day needed to accomplish FW conditions for each hour between 0 and 23 UT (Kubicki et al., 2016). This restriction limited the amount of available data. Michnowski et al. (2021) used FW PG values with at least 12 hours of fair weather conditions. In this work we increase the amount of data taking in account every FW hourly value to calculate the diurnal and seasonal variations.

### 4.1  Diurnal variation of PG

Fig. 5 presents the PG diurnal variation by season (winter – December, January, February, spring – March, April, May, summer – June, July, August, autumn – September, October, November) and the all year mean variation. The shape of the PG curve is substantially different depending on the season. During the winter months there is a single broad maximum lasting from the

noon (∼13 UT) until late evening (∼20 UT) whereas during the spring-summer-autumn period double maxima (∼7 and ∼19 UT) are present. The amplitude variation of the PG curve during the winter is larger than in the summer (Tab. 2). Świder is thus characterised by the type I of the "double oscillation" variation in the diurnal average curve as according to Israël (1973a), i.e. the double oscillation is prevalent except in local wintertime and its maxima occur at different hours depending on the season. Differences between the PG diurnal curves at Świder and the classical Carnegie diurnal curve come from the much greater variability and dynamics of the planetary boundary layer (PBL) characteristic for the location and at Świder affected by pollution, meteorological factors and seasonal effects (Kubicki et al., 2016; Nicoll et al., 2019; Michnowski et al., 2021, Section 4).

Since we cannot calculate the whole diurnal variation of CN concentration we show results only for 6, 12, 18 UT (represented by symbols) in Fig. 6. On the background (solid lines) are shown the Świder 1968/69 diurnal curves from Warzecha (1976) obtained by the recordings with the Verzár counter every 15 minutes. We note that in this comparison we calculated the mean values only for 24-hour FW days since Warzecha presented the diurnal curves for cloudless days. The highest mean CN concentrations at these hours are registered at 12 UT although looking at diurnal curves of Warzecha (1976) this is already after the morning maximum of CN concentrations between 6 and 12 UT (all seasons) and before the afternoon/evening maximum. These values are especially high in the winter and the spring. The concentration values at 6 and 18 UT are similar, however the lowest values are noted in the morning. At 6 and 18 UT the mean CN concentrations in the spring and autumn are almost equal while at 12 UT the spring mean is higher, even than the winter value which generally is the highest among seasons. Compared with the diurnal curves of Warzecha (1976) the CN concentrations fairly agree with the diurnal variation except for the 18 UT (excluding summer) and for 12 UT in summer and autumn. Some differences are likely since we compare long-term mean values with one year means. Nevertheless, it is worth noting that the diurnal CN curves fall between the CN averages at one standard deviation.

## 4.2 Seasonal and annual variation of PG

Analogically to Fig. 4 in Fig. 7 we show the seasonal differences in the distribution of FW PG at different CN concentrations. Their main statistical parameters for seasonal distributions are given in the figures as well as in Tab. 2. Fig. 7 presents the seasonal histograms of the hourly averaged (at 6, 12, 18 UT) FW PG values for 1965–2005. Each histogram contains two types of distributions: red colour indicates the FW PG values measured simultaneously with CN concentrations, and green regions with black contour indicate a subset of the FW PG values for CN concentrations less than 10000 cm$^{-3}$. The main statistical measures for both of these distributions are displayed within each plot. Depending on the season we observe different PG distribution characteristics. For spring, summer and autumn most values ranged between 100-300 V/m with mean value of ∼250 V/m in the spring and autumn and ∼200 V/m in the summer. Winter values ranged between 200–400 V/m and the mean was ∼350 V/m. The standard deviation is also the highest in the winter. The kurtosis values for the spring and autumn are 4.7-4.8 while the summer kurtosis is the highest at 5.7 and the winter is the lowest at 3.2. The skewness of the winter distribution is the lowest of 0.6, and the skewness of all the other distributions is between 0.8-1.2 indicating more values concentrated around the mean value. We have used the non-parametric test of Kolmogorov-Smirnov (KS) and Mann-Whitney U (MW)

test to estimate the statistical significance of the difference between the PG distributions for decreasing CN concentrations. In regard to the distributions shown in Fig. 2 and in Fig. 7, both KS and MW tests indicated statistically significant differences (at the significance level of 0.05) between the PG-CN all and PG-CN<10000 distributions for the whole year as well as in the spring, autumn and winter. In case of the variations shown in Fig. 11, the KS test indicated statistically significant differences between the PG populations for all CNs and CN<10000, in January and February only. The MW test indicated such in April and November as well. When comparing PG and CN all with PG and CN<8000 the results of both tests indicate the statistically significant difference also in December, similarly to PG and CN<6000.

Another way to look at the relationship between aerosol or condensation nuclei concentration is presented in the Fig. 8. Intensity of colour indicates the number (counts) of data points at certain range of PG and CN. Here the PG values are plotted against CN concentrations by season. In the summer there seems to be little influence of CN concentration on PG as the similar range of PG values is observed at wide range of CN concentrations. Some kind of dependency is visible in the winter, and spring and autumn are intermediate in this sense with some high PG values at high CN concentrations.

## 5   Diurnal, seasonal and annual variation at different level of CN concentration

When the continuity of the conduction current is fulfilled, the atmospheric potential gradient depends on local conductivity of the air. The process of attaching of small ions to aerosol particles causes reduction in the ion concentration (loss of small ions) and change in their mobility as large ions and, consequently, a decrease of the electrical conductivity and an increase of the PG values (e.g., Israël, 1973a; Bennett and Harrison, 2008; Matthews et al., 2019; Wright et al., 2020). In this section we analyse if limiting CN concentrations to the possible lowest ranges in the CN distribution affects the PG values and its variations. Firstly, we consider CN concentrations below 10000 $cm^{-3}$ (black-green histograms in Fig. 4) and next decrease the concentration limit by 2000 $cm^{-3}$ to 8000 $cm^{-3}$, and next to 6000 $cm^{-3}$, to investigate the diurnal, seasonal and annual variations of PG at these lowest CN concentrations occurring at Świder. In earlier sections we showed that limiting the CN concentration to 10000 $cm^{-3}$ did not change significantly the kurtosis and skewness of the PG distributions, however their average mean values differ (see also Sec. 7), and here we investigate if any possible extreme CN limits affect variations of the PG in this representation. The right part of Tab. 1 presents the counts and percentages of the whole fair-weather PG population of values within each CN concentration limit. In Fig. 9, in the upper panel we show the histograms of CN concentrations and in next panels are shown the histograms of PG values at the three selected limits of CN concentration. There are 3736 values below 10000 $cm^{-3}$, 2086 values below 8000 $cm^{-3}$ and 939 values below 6000 $cm^{-3}$, which is about 23%, 13%, 6% of all CN concentrations, respectively. Below 4000 $cm^{-3}$ there was only 123 values (0.8%) and therefore this subset was not taken into account. These values represent the whole span of the studied time period, as shown in the small panel in the top panel of Fig. 9. They also represent all seasons as shown in Tab. 6, even though there is proportionally more counts in the warmer season because of more frequent FW conditions. In the lower panels of Fig. 9 the histograms of corresponding PG values are shown. We note a slightly decreasing tendency in the mean value of the population especially at CN concentrations below 6000

**Table 3.** Mean values of fair weather PG by season for all and limited CN concentration values together with the one standard error of the mean corresponding to 6, 12, 18 UT and on the average.

| | Winter | | | | Spring | | | |
|---|---|---|---|---|---|---|---|---|
| Hour | 6 UT | 12 UT | 18 UT | Mean | 6 UT | 12 UT | 18 UT | Mean |
| CN all | $291 \pm 6$ | $370 \pm 5$ | $373 \pm 6$ | $351 \pm 3$ | $251 \pm 3$ | $227 \pm 3$ | $264 \pm 3$ | $250 \pm 2$ |
| CN <10000 | $265 \pm 9$ | $339 \pm 12$ | $320 \pm 10$ | $305 \pm 6$ | $228 \pm 5$ | $217 \pm 6$ | $238 \pm 7$ | $228 \pm 4$ |
| CN <8000 | $258 \pm 12$ | $333 \pm 16$ | $318 \pm 13$ | $299 \pm 8$ | $231 \pm 7$ | $219 \pm 8$ | $228 \pm 11$ | $227 \pm 5$ |
| CN <6000 | $225 \pm 15$ | $342 \pm 19$ | $316 \pm 23$ | $278 \pm 12$ | $220 \pm 9$ | $215 \pm 9$ | $228 \pm 20$ | $220 \pm 6$ |

| | Autumn | | | | Summer | | | |
|---|---|---|---|---|---|---|---|---|
| Hour | 6 UT | 12 UT | 18 UT | Mean | 6 UT | 12 UT | 18 UT | Mean |
| CN all | $227 \pm 4$ | $253 \pm 3$ | $272 \pm 4$ | $253 \pm 2$ | $215 \pm 2$ | $169 \pm 2$ | $187 \pm 2$ | $193 \pm 1$ |
| CN <10000 | $210 \pm 8$ | $238 \pm 6$ | $261 \pm 10$ | $235 \pm 4$ | $207 \pm 3$ | $168 \pm 3$ | $188 \pm 3$ | $192 \pm 2$ |
| CN <8000 | $213 \pm 10$ | $231 \pm 7$ | $256 \pm 13$ | $231 \pm 5$ | $208 \pm 4$ | $169 \pm 4$ | $194 \pm 4$ | $193 \pm 2$ |
| CN <6000 | $208 \pm 12$ | $227 \pm 9$ | $242 \pm 19$ | $224 \pm 7$ | $202 \pm 7$ | $174 \pm 5$ | $196 \pm 6$ | $190 \pm 3$ |

$cm^{-3}$ approximately 210 V/m and lower kurtosis and skewness compared with the distributions below 10000 $cm^{-3}$ and 8000 $cm^{-3}$.

Seasonal means are shown in Tab. 3 and Fig. 10 for each hour term (6, 12, 18 UT) and the mean average over the terms, together with the standard error of the mean. Here we also notice a decrease in the mean values of PG for each term at lower CN concentrations particularly at 6000 $cm^{-3}$ limit (i.e., comparing PG values between CN-all and CN below 6000 $cm^{-3}$). This is present in the winter ($73 \pm 15$ V/m or $\sim$13% for total mean value), spring ($30 \pm 8$ V/m or $\sim$12%) and autumn ($29 \pm 9$ V/m or $\sim$12%) and the decrease are larger for 6 UT (e.g., $30 \pm 12$ V/m or 12% in the spring) and 18 UT (e.g., $57 \pm 29$ V/m or 15% in the winter, $36 \pm 23$ V/m or 14% in the spring). However, in the summer there is practically no effect of CN and no clear tendency of decrease or increase (the largest decrease of several V/m only).

Finally, in Fig. 11 and Tab. 4 the annual variation of FW PG is presented, calculated as monthly averages. Additionally in Table B3 in the Appendix B the number of FW PG values calculated as monthly values for all and limited CN concentration is presented. PG variations calculated at 6, 12, 18 UT are plotted separately from the average curve. The maximum of the PG is in February which is also the maximum at 12, 18 UT. The minimum of PG is in June which is also the month of the minimum at 6, 12 and 18 UT. The lowest PG amplitude of the differences between the maximum and the minimum are in the 6 UT curve: $91 \pm 12$ V/m (wide maximum Dec-Mar, minimum in June) and much higher PG differences are at 12 and 18 UT: $213 \pm 12$ V/m (Feb-Jun) and $217 \pm 13$ V/m (Feb-Jun/Jul), respectively, $174 \pm 7$ V/m (Feb-Jun) in the average curve.

In the PG variations represented by the curves calculated with limitations of CN concentration there is a significant reduction of PG from October to March (11%–26%) and the biggest differences are seen in February at each term 6, 12, 18 UT and on

average. In February the PG decreases by 95±25 V/m or 26%. In January and November the change is about 20% (75±24 V/m and 58±20 V/m, respectively) and in December about 13% (45±28 V/m). From April to September there is a lack of or little influence of low CN concentration, and the PG differences are mostly of several V/m, which is of the order of the standard deviation of the mean error, except for April and September where these differences can be larger than 10 V/m. In September at 18 UT we note a larger decrease of PG of 44±29 V/m (18%). In addition, as we already have seen in the seasonal plots in Fig. 10 at 12 UT there is even an inverse relationship between the PG and CN. For example in July at 12, 18 UT and in August at 18 UT, although these differences are within the statistical error. In terms of the change of the amplitude of maximum–minimum differences at 6 UT there is a small change with decreasing CN concentration reaching 77±41 V/m (Dec–Jun) at 6000 cm$^{-3}$ CN limit. At 12 UT it is 165±35 V/m (Jan–Jul) at 6000 CN limit while at 10000 and 8000 CN limit it is practically the same as with no limits (∼213±12 V/m, Jan/Feb–Jun–Jul). At 18 UT at all CN limits the amplitude is ∼160 V/m compared with 217±13 V/m with no limits. There seems to be slight shift between the maximum or minimum month which is January or February for the maximum, and June and July for the minimum. In general, there are significantly lower differences between winter and summer in the annual variation of the FW PG at the lowest CN concentration limit considered, however, the maximum still stays in the winter (Dec–Jan–Feb).

## 6   PG and conductivity model with variable aerosol content

In this section we model a hypothetical change in the PG when we limit the number concentration of CN. The relationship between atmospheric electric potential gradient and aerosol concentration is complex. These two most often measured variables are related through the electrical conductivity of the air. The conductivity, $\sigma$, is influenced by the loss of small atmospheric ions due to attachment to aerosol particles, and decreases when the attachment intensifies. When we assume that the mobility and concentration of positive and negative ions are equal, and the ions are singly charged, then, in a steady state the ion conductivity is determined by Eq. 1 (e.g. Schonland, 1953; Makino and Ogawa, 1985; Sapkota and Varshneya, 1990).

$$\sigma = 2e\mu n = \frac{4e\mu q}{\beta N + \sqrt{(\beta N)^2 + 4\alpha q}} \tag{1}$$

where $e = 1.6 \times 10^{-19}$ C is the elementary electric charge, $\mu$ is the ion mobility, and $n$ is the ion concentration, $q$ - the ion production, $\alpha$ is the ion recombination rate coefficient, and $\beta N$ represents the ion loss due to attachment to particles and droplets of concentration $N$, and $\beta$ the attachment rate coefficient. In the atmosphere the loss of ions in fair weather is mainly due to the attachment to aerosol particles. The process of attachment is dependent both on the number concentration and the spectrum of the size of aerosol particles, and is usually expressed by the ion-aerosol loss term, in general being a sum of contributions over different aerosol types and sizes (e.g., Israël, 1973a; Tinsley and Zhou, 2006). The term can be replaced by the effective coefficient $\beta$ and the total aerosol concentration $N$, i.e. $\beta N$ as in Eq. 1. We set the other parameters in Eq. 1 as follows: $q = 2.2$ cm$^3$s$^{-1}$, $\alpha = 1.3 \times 10^{-6}$ cm$^{-3}$, $\mu = 1.5$ cm$^2$Vs$^{-1}$, similarly to other conductivity models based on other empirical models and measurements (Makino and Ogawa, 1985; Sapkota and Varshneya, 1990; Tinsley and Zhou,

2006; Kulkarni, 2022). The ion production by cosmic rays of the order of $1.0 \, \text{s}^{-1} \, \text{cm}^{-3}$ is appropriate for the production at the ground level, and of the order of $1.0 \, \text{s}^{-1} \, \text{cm}^{-3}$ for the production by radioactivity of radon. We set the total ion production value at Świder to $2.2 \, \text{s}^{-1} \, \text{cm}^{-3}$. The mobility of $1.5 \pm 0.3 \, \text{cm}^2 \, \text{s}^{-1} \, \text{V}^{-1}$ up to 15 km could be used for both positive and negative ions according to (Swider, 1988).

In case where the conduction current, $j_C = \sigma \, \text{PG}$, is preserved, a decrease or increase in the electrical conductivity should

result in a proportional increase or, respectively, decrease in the potential gradient:

$$\frac{\sigma_2}{\sigma_1} = \frac{PG_1}{PG_2} = s \qquad (2)$$

where indices 1 and 2 refer to value before and after the change, respectively.

Rural continental aerosols of natural origin occur in Świder but there may be a lot of pollution from suburban traffic and

domestic heating (Majewski and Przewoźniczuk, 2009; Pietruczuk and Jarosławski, 2013). More recent observations to the considered CN concentration measurements at Świder, of aerosol size distribution, confirm the occurrence of polluted aerosol which consists of large concentrations of fine particles dominated by the nuclei mode of the mean diameter 20–30 nm, and by the accumulated mode of the mean diameter 100–120 nm (Kubicki et al., 2016). We want to focus on the winter conditions when the decrease of the potential gradient at lower CN concentrations, expected to be due to decreasing effect of pollution on

the air conductivity, is the biggest as obtained in Sec. 4, and next, to estimate the effect during the summer. We will emulate the conditions by calculating the conductivity at different aerosol composition and concentrations. Let us consider a mix of continental aerosol types the Świder aerosol may be composed of, and calculate the corresponding conductivities as well as the ratio of the conductivities as in Eq. 2.

According to Hess et al. (1998) a land aerosol may consist of 3-4 main aerosol types: continental clean, continental average,

continental polluted, and urban. These aerosol types consist of three basic aerosol components: water insoluble *(WatInsol)*, water soluble *(WatSol)* and soot *(Soot)*, the two latest contributing almost 99% of the total concentration and contained in the range of CN particle size. Their individual $\beta$ coefficients are: $\beta_{WatInsol} = 2.80 \times 10^{-5} \, \text{cm}^3/\text{s}$, $\beta_{WatSol} = 1.18 \times 10^{-6} \, \text{cm}^3/\text{s}$, and $\beta_{Soot} = 5.56 \times 10^{-7} \, \text{cm}^3/\text{s}$, respectively, obtained from calculations based on the equations used in Tinsley and Zhou (2006). In this model the effect of relative humidity on the conductivity is not included.

$$\beta N = \beta_{WatSol} N_{WatSol} + \beta_{WatInsol} N_{WatInsol} + \beta_{Soot} N_{Soot} \qquad (3)$$

where $N_{WatInsol}$, $N_{WatSol}$, and $N_{Soot}$ are the number concentrations of the three aerosol components.

At first we set the limiting values of concentration $N$ for 100% relative composition of soot versus 100% water soluble aerosol at $26000 \, \text{cm}^{-3}$ ($N_{Soot}$) and $6000 \, \text{cm}^{-3}$ ($N_{WatSol}$), respectively. The first value corresponds to the highest average

aerosol concentrations in the winter (Fig. 4, Tab. 2), and the second value refers to the lowest concentrations measured at Świder

($\sim$4000 occur too, but they are very rare). A small, constant contribution of insoluble aerosol of 500 cm$^{-3}$ is also assumed. In Tab. 5 we present the results of the model calculations when the relative proportions of water soluble and soot components vary (columns 1-2), but the total concentration is dominated by the soot component, except at the lowest ratios of $N_{Soot}/N_{WatSol}$, which result in water soluble aerosol at a low number concentration. Columns 3-5 give the resulting number concentrations of $N_{Soot}$, $N_{WatSol}$, and the total concentration CN. In the column 6 we give the calculated ion-aerosol loss term $\beta N$, and in the column 7 the effective ion attachment rate $\beta_{eff} = (\beta_{eff}N)/N$ which here varies from 1.07 to 3.24 $\times 10^{-6}$ cm$^3$ s$^{-1}$. Sapkota and Varshneya (1990) mention that Hoppel predicts a range of 0.8 to 3.0 $\times 10^{-6}$ of $\beta_{eff}$ for the continental aerosol. In the last column we give the calculated conductivity. Twice the polar (positive) conductivity calculated from newly digitised 1965–2005 data: $4.4 \pm 0.2 \times 10^{-15}$ S/m for the winter, and $8.0 \pm 0.2 \times 10^{-15}$ S/m for the summer. The value given equals twice the seasonal average of the positive polar conductivity calculated from the hourly values reported in the observatory yearbooks (in fair weather conditions).

An average winter situation may correspond to the concentrations of 1200 cm$^{-3}$ of water soluble aerosol and 18200 cm$^{-3}$ of soot which give the total CN concentration of 20500 cm$^{-3}$, since the average observed mean is $\sim$20900 cm$^{-3}$. For the observed median, $\sim$18200 cm$^{-3}$, 1800 cm$^{-3}$ of water soluble aerosol and 15600 cm$^{-3}$ of soot could be more appropriate. The conductivity in these cases, 4.56 and 4.69 $\times 10^{-15}$ S/m, are close to $4.4 \times 10^{-15}$ S, an average observational winter value. This has been adjusted by assuming $q = 2.2$ cm$^3 s^{-1}$. When we look for concentration characteristic for the summer it would rather be reflected by 4200 cm$^{-3}$ of water soluble aerosol and 7800 cm$^{-3}$ of soot for the summer average mean $\sim$15300 cm$^{-3}$, and the median $\sim$12300 cm$^{-3}$. The total CN concentration is 14500–12500 cm$^{-3}$, since the observational (Fig. 4, Tab. 2). However, the conductivity for such concentration is $4.97 \times 10^{-15}$ S/m and remains much too low compared with the observations, since the summer average is $\sim 8.0 \times 10^{-15}$ S/m. Such conductivity level requires the ion-aerosol loss term $\beta N$ be about $1.5 \times 10^{-2}$ s$^{-1}$, and the effective $\beta_{eff} = 1.25 \times 10^{-6}$ at $N = 12000$ cm$^{-3}$ or $\beta_{eff} = 1.0 \times 10^{-6}$ at $N = 15000$ cm$^{-3}$. Such values of $\beta$ are more of the order of $\beta_{WatSol}$, therefore we now consider a consider a mix of the three selected aerosol components dominated by the water soluble component. We set the limiting values of concentration $N$ for 100% relative composition of soot versus 100% water soluble aerosol at 6000 cm$^{-3}$ ($N_{Soot}$) and 28000 cm$^{-3}$ ($N_{WatSol}$), respectively. Now the lowest concentrations are only due to soot, and small additions of water soluble concentrations. A small, constant contribution of insoluble aerosol of 200 cm$^{-3}$ is assumed in this case. The results of the conductivity calculations are shown in Tab. 6.

An average summer situation may correspond to the concentration ratio of 40% vs 60% of the water soluble and soot components which give the total CN concentration of 15000 cm$^{-3}$. The conductivity in this case is $5.73 \times 10^{-15}$ S/m. When we look for concentrations characteristic for the winter they would rather be reflected by the ratio of 70% vs 30%, i.e. CN total 21600 cm$^{-3}$, and the model conductivity is close to the observational value of $\sim 4.4 \times 10^{-15}$ S/m. It seems this conductivity model could describe both winter and summer conditions, however situations from the model of Tab. 5 cannot be excluded.

There may be other atmospheric particles that cause a larger difference between the winter and summer conductivities, like the dust particles, even though these are in general much less numerous than CN. The conductivity value very much depends on both $\beta_{eff}N$ and $q$, with $\beta_{eff}N$ depending on the distribution of the sizes of the aerosol particles. In particular, the insoluble component also plays an important part through the high attachment rate. These may also vary between the summer

and the winter. More analysis, and observational data from Świder are needed to develop a more realistic conductivity model, particularly of the aerosol size distributions.

## 7 Effect of low CN concentrations on the air positive conductivity

In this section we analyse the annual variation of the fair weather positive conductivity (PC or $\lambda_+$) and the effect of decreased CN concentrations, similarly as in Sec. 5. The total number of FW PC hourly values is lower than the total number of FW PG: 124213 hourly values (about 95% of the PG set), and 15187 values (about 12% of all FW PC) have the corresponding hourly value of CN concentration at 6, 12 or 18 UT. The maximum of the PC is between June-July, which is also the maximum for 18 UT. At 6 UT and 12 UT, the PC has a wide maximum between May-August. The minimum of PC is minimum in February, which is also the month of minimum at 6, 12 and 18 UT. There are also data sets of considerable size for the calculation of the annual variation which we present in Fig. 12. The all average mean of the conductivity is $3.28 \pm 0.01$ fS/m, as indicated in the first row of Tab. 7. There we give also seasonal values with their standard errors. The winter positive conductivity $2.24 \pm 0.01$ fS/m is 1.8 times the highest seasonal, summer conductivity of $3.98 \pm 0.01$ fS/m. Contrary to the effect on PG, for each season we observe an increase in the mean values of PC at lower CN concentrations, including the summer - unlike with the PG. The biggest differences were noted when comparing PC values for all CN and CN below 6000 cm$^{-3}$ (summer, winter and the whole year), or CN below 8000 cm$^{-3}$ (in the spring and autumn). An increase of about 17% was noted for the spring (0.58 fS/m), the autumn (0.52 fS/m) and for the winter (0.36 fS/m). The lowest increase of about 8% was observed in the summer (0.33 fS/m), however this still more than the corresponding change in the PG. Additionally, we noted a change of 32, 34, 37, 38, and 36 % in the winter and 21, 27, 15, 14, and 15 %, respectively in the summer, when comparing the winter/summer PC variation with annual averages for all values, CN all, C<10000, C<8000, and C<6000, respectively. From Tab.2, we found that for winter (summer) there is a 52% (16%) difference in the PG values, and 10 % (16%) in total CN concentration. This is more evident in the Fig. 14 where we plot the annual variations of the departures of the PG, the total conductivity (TC) and concentrations of CN from the annual mean. Therefore, we can conclude that removing the CN, even down to quite lower concentration, is not completely writing off local influences that could be affecting the PC (and therefore the PG).

The effect of lower CN concentrations could also be considered by season, as we discussed in Sec. 5. We extend our analysis, and in Fig. 13 we summarise the change in seasonal averages of the Świder FW PG and TC at the low CN concentration considered during the studied period, and compared with other periods 1957-1964 and 2005-2015, for which we have gathered positive and negative conductivity data. Data from 1957-1964 come from additional Observatory yearbooks (see App. A) and data from 2005-2015 have been calculated from digital datasets collected during other project (Odzimek et al., 2018). However, up to 1964 the conductivity was only evaluated at the three terms 6, 12 and 18 UT, similarly to the measurement schedule for CN. In 2005-2015 there is a similar effect on the conductivity as it is in 1965-2005 but the period 1957-1964 differs from the later periods in an effect on PG present also during the summer. The atmospheric electricity during that era was affected in the long-term by the effect of nuclear weapons tests, and due to more radioactive material present throughout the atmosphere effect on the electric variables could be different, and spread differently by season (this requires more study).

Analysis of data from 2005-2015 resembles results based on potential gradient and positive conductivity from 1965-2005. The amplitudes of absolute changes and relative changes differ between the periods, however, indicating rather complicated nature of the relationships, probably depending most on the composition and types of aerosols as well as activity of ionising agents. Looking at the differences between the effects on the potential gradient and conductivity, we conclude that while there some effect of conductivity present in all seasons, the effect on summer PG is practically absent. It further confirms there are some non-Ohmic contributions to PG, and therefore the annual variation of PG at such site are difficult to interprete, including any connections with annual variability of the GEC. These issues need to be further investigated.

One way of the removal of the effects of the mid-day convection is to consider the night-time conditions, or selecting periods of more stable conditions. Another issue is the effect of other aerosol types like dust particles which are not all measured by CN counters and affecting the conductivity and indirectly the potential gradient which needs to be taken into account. One may consider if a similar procedure with threshold values of the conductivity and corresponding effects in the PG annual variation. This is presented in Fig. 13 (left panel) where, similarly to Fig. 13, we plotted the annual variation of PG at values of the total conductivity higher than 3.0, 5.0, 8.0 fS/m. The biggest effect is observed during winter, and a very small one again in the summer. This confirms the winter pollution strongly affects the PG, which may come from the load of fractions of aerosol and dust particles, outside the range of measured condensation nuclei. In Fig. 15b we present the annual variability of PM10 mass concentrations at Świder from the period 2005-2015. The highest PM10 mass concentrations were observed in the period from October to April with a maximum values of 62.1 $\mu$g/m$^3$ and 51.2 $\mu$g/m$^3$ noted in February and March, respectively. The average values of PM10 in October, November, December, January and April are comparable and ranged between 43.5 $\mu$g/m$^3$ and 47.3 $\mu$g/m$^3$. The smallest concentrations were observed from May to September with the lowest value (20.9 $\mu$g/m$^3$) noted in June. The mean values of PM10 in the period: May-September (except in June) oscillated between 23.0–28.7 $\mu$g/m$^3$. Such annual variation is characterised for other stations located in Central Poland (Pietruczuk and Jarosławski, 2013). Elevated values of PM10 registered during late autumn, winter and at the beginning of spring related to more intensive domestic heating during cold part of the year. Additionally specific meteorological conditions observed during winter anticyclones (low wind speed and stable boundary layer) are favorable for accumulation of PM10. The enhanced emission of dust, particularly in the winter, may have strong effect on the conductivity. On the other had, with the levels of mass concentration of PM10 in the summer, even though having much smaller variability (lower standard deviation), we do not observe much effect on the PG then. Again, this may indicate a non-ohmic contribution to the PG present notably in the summer. We also need to note that the variations shown in Fig. 15a and Fig. 15b are difficult to compare and draw conclusions because what is measured as PM10 is the mass concentration, and for conductivity we require number concentrations and particle size spectra.

## 8 Discussion

At many land atmospheric electricity stations, including Świder, the increase of the potential gradient during local wintertime or rush hours, which is usually related to increase of emission from intensive domestic heating and transport, is considered to be due to a decrease in the electrical conductivity (Harrison, 2006; Kubicki et al., 2016; Tacza et al., 2021). Having at disposal

all digitised PG hourly values we calculated the diurnal variation of PG from all available fair weather hours. As in the previous results of such analysis at Świder (e.g., Warzecha, 1991; Kubicki et al., 2007), made on the basis of 24-hour fair weather days 1965-2000, or fair weather hours during days with 12 hours of FW conditions over the time period 2004–2011 (Michnowski

et al., 2021) the PG diurnal variation shown in Fig. 5 confirmed the different behaviour according to season. The double oscillation is evident during spring, summer and autumn (in the summer with peaks at 7 and 20 UT), and in winter a single oscillation (a minimum at ∼3 UT and a peak at 19 UT). Similar PG diurnal curves were found in Reading for summer and winter (Nicoll et al., 2019). One hypothesis for the two PG peaks found in summer at urban sites could be due to pollution of vehicular traffic during rush hours. Majewski and Przewoźniczuk (2009) performed Particulate Matter (10$\mu$m) measurements

for eleven stations in Warsaw. They found clear two peaks (7 and 20 UT) in warm weather (April to September) and less pronounced peaks in cold weather (October to March). As mentioned earlier, Świder is located in a suburban site so vehicular transportation is not expected affect too much in the PG diurnal variation. Then, the hypothesis for the difference in the PG diurnal variation in summer and winter is a combination of "sunrise effect", generally thought to be related to mixing of the near-surface electrode layer (which is an accumulation of positive charge next to the negatively charged Earth's surface). The

first peak (at ∼7 UT) could be associated both with the dynamic changes in the planetary boundary layer and generation of the secondary aerosol (Kubicki et al., 2007). Then, as the temperature is continuously increasing, the convection intensifies producing mixing processes in the PBL causing transport of aerosol to higher altitudes, and therefore the PG decreases. As the temperature decrease after 16 UT the PG return to normal values. In winter, the reduced variability in PBL height (due to diminished convection) therefore leads to more quiescent meteorological conditions which results in a more stable diurnal

variation in PG, and the disappearance of the morning maximum peak (see also Nicoll et al., 2019).

The annual cycle of emissions with maxima in the winter and minima in the summer affecting the land PG measurements (Adlerman and Williams, 1996; Nicoll et al., 2019; Shatalina et al., 2019) are considerable at Świder, as confirmed in the analysis in Sec. 3 (Figs. 5, 6, 8). Similar behavior in the PG seasonal variation was found in Kew and Reading, in the UK, and Nagycenk, in Hungary (Märcz et al., 1997; Harrison and Aplin, 2002; Nicoll et al., 2019). Harrison and Aplin (2002)

associated the PG higher values during winter, at Kew, to the influence of smoke pollution. Märcz et al. (1997) suggested that the PG seasonal variation at Nagycenk is likely associated to the condensation nuclei variation. Nicoll et al. (2019) suggested that the PG higher values during winter at Reading is due to more use of domestic heating, and it is very likely that at Świder the PG values are higher during winter due to domestic heating producing soot. This is also in agreement with the simple conductivity model in this work. Measurements of PM$_{2.5}$ in Krakow (south of Poland) in 2020/2021 found higher PM$_{2.5}$

concentrations during winter month compared with summer months (Ryś and Samek, 2022). The time intervals of this study and the PG measurement at Świder are different (in fact both site locations are very far away), however, a similar behavior in PM is very likely at Świder. The aerosol concentration changes also annually due to variation in convection processes in the planetary boundary layer (PBL), and the PBL height, more influential during warm season and especially in the local summertime. At Świder, as discussed in Sec. 3, the summer CN concentrations differ much more from the rest of the year

(Fig. 4) being minimal through all the year. As opposed to winter the change in the summer concentrations does not seem to be so significant on average (Fig. 9). The effect of reduced aerosol concentration to its lowest levels did not change the

average PG amplitude in the summer, while in the other seasons this effect was observed particularly in the winter (Sec. 4). However, analysis of annual variations of PM10 mass concentrations measured at Świder in 2005-2015 indicate twice stronger concentrations and high variability in the winter. It is difficult to verify quantitatively the effect it has on the air conductivity but the analysis shown in Fig. 15 indicates some qualitative effects, especially in the winter, most possibly due to increased load of dust.

Dust particles used to be measured at Świder observatory, using different methods. Modern measurements are performed in collaboration with Poland's state agencies such as the State Sanitary Inspection or Chief Inspectorate for Environmental Protection. At the beginning an Owens dust counter was used (Kalinowska, 1962). We are not aware of any digital records from this period. Using the measurements with the counter from 1957-1960, and the measured condensation nuclei concentrations, Haberka (1961)[1] investigated the dependence of the amount of the dust particles compared to the concentration of CN. At concentrations higher than $10000 \ \mathrm{cm}^{-3}$ the number of dust particles was larger than 37. In the CN range 5000-10000 $\mathrm{cm}^{-3}$ it was 27 in one $\mathrm{cm}^3$, a value relevant to dust concentration in warm season 24-29 per $\mathrm{cm}^3$ (48-70 per $\mathrm{cm}^3$ in cold the season). This result supports method presented in this study, however, the relationships between aerosol, dust, air conductivity and the atmospheric potential gradient are complicated, and the lack of thorough and continuous measurements of all relevant quantities further complicates such a study.

The changes of PG due to changes in the conductivity and other processes obscure the real changes of the PG resulting from the activity of the GEC at a mid-latitude land station like Świder, and the annual maximum of PG occurs usually during local winter. In this work we wanted to investigate in what way taking into account fair weather PG values at low CN concentrations affects the PG annual maximum. In general, our results show that even though the PG decreases at low CN concentrations possible in the winter and in the autumn, the maximum of PG remains in the winter, and when investigating similar effects on the conductivity they are differently dependent on season (Tab. 7, Fig. 12, Fig. 13). When looking at the annual curves of variation of the PG, total conductivity (TC) and concentrations of condensation nuclei these have differing departures from the annual mean, and these changes are caused by different factors. The changes in the annual or seasonal course of PG are not adequate to changes in TC or CN concentrations, and neither in PM mass concentrations (Fig. 13–15).

According to the conductivity model for varying ground-level CN concentrations characteristic for Świder, described in Sec. 5, there could be increases in the ion conductivity, and decreases of the PG, but the relative change differs from the observed conditions. This requires further investigation. Analysis of conduction current density as product of PG and conductivity may provide more insight into the annual variation, however, the effects of any non-ohmic contribution such as due to convection will still be difficult to remove.

## 9 Summary

The findings of this work could be summarised as follows:

---

[1] The correct surname is most likely Haberko.

- We calculated diurnal, seasonal and annual variations of fair weather potential gradient based on 1965–2005 digitized time series of data from Świder observatory yearbooks, and we provided details of the statistics for the PG and the condensation nuclei CN number concentrations, measured during this period at three specific hours of measurement (6, 12 and 18 UT). The general average mean of PG is 230 V/m, and the average PG at the CN concentration observation terms is 248 V/m while the CN average mean is about 19000 $cm^{-3}$. The PG and CN distributions are both leptokurtic (i.e., narrow) and right-skewed. A subset of PG values for CN concentration below 10000 $cm^{-3}$ (which correspond to 20% of the whole CN-PG pair data) was selected for further analysis of the effect of aerosol concentration on the PG variation. In particular, we aimed to investigate the PG annual variation in connection with the GEC variability. The PG average mean for reduced levels of CN concentrations is 221 V/m, and the CN average mean in this subset is 7290 $cm^{-3}$.

- Summer and winter months have a clear signature in the PG diurnal variation for all CN concentrations, likely associated with mixing processes in the planetary boundary layer. For the subset with CN below 10000 $cm^{-3}$, we found that PG values do not show any significant variation in the summer. On the other hand, for the winter we found a PG difference of 11%. Furthermore, this PG difference is about 20% when we consider CN below 6000 $cm^{-3}$.

- Local winter months have higher PG values compared with local summer months, a common feature of continental atmospheric electricity stations. This PG seasonal variation is maintained even at lower CN concentrations (e.g., 8000, 6000 $cm^{-3}$). We found that there is a reduction in the PG values by 11-26% with the biggest difference in February. In the summer months this difference is negligible. From this, we can conclude that aerosol at these concentrations has more effect in the PG during winter compared with summer.

- On the opposite, local summer months have higher positive conductivity values compared with local winter months, for Świder there is factor of 1.8 between the corresponding mean average values $3.98 \pm 0.01 \times 10^{-15}$ S/m and $2.24 \pm 0.01 \times 10^{-15}$ S/m, respectively. At lower CN concentrations (e.g., 10000, 8000, 6000 $cm^{-3}$) we found an increase in the conductivity by 7–17% over the winter, spring and autumn and summer. This is different from the effect on the PG, and also could indicate a non-ohmic PG component or a component which is not very sensitive to CN concentration levels, or both. Any future analyzes of the annual variation of the PG in the hope of finding an annual variation of the GEC have to take these effects into account.

- Analysis of additional datasets from 2005-2015 confirm similar influence of reduced CN on PG and the conductivity. A question remains whether any PG data from a location like Swider could be used for the investigation of the annual variation of the GEC. There are issue like the effects of the mid-day convection. Another issue is the effect of other aerosol types like dust particles which may not all be measured by CN counters. They affect the conductivity and indirectly the potential gradient which needs to be taken into account. These methods both require continuous or more frequent monitoring of aerosols. Analysis of the annual variation of the conduction current is another possibility of investigating these variations, however, difficulties in removing the effects of any non-ohmic contribution such this due

to convection will still be difficult to overcome. Simultaneous measurements of Maxwell current density may also be required in such case (Odzimek et al., 2018).

– A simplified modelling of electrical conductivity affecting the PG is created. The ion loss in this model is due to attachment mainly to water soluble aerosol particles and soot particles, the main components of continental aerosol. A more realistic model should include effects on conductivity of dust or soot of the particle size larger than the CN.

*Data availability.* Świder atmospheric electricity and condensation nuclei data are available on request from co-author A. Odzimek (aodzimek@igf.edu.pl). Świder PM10 data 2005-2015 were downloaded from the data archives of the Chief Inspectorate for Environmental Protection, Poland.

## Appendix A

This Appendix contains the list of Świder Observatory yearbooks used in this study.

Yearbooks including data from 1957-1964

Warzecha S. (Ed.), 1960: Rocznik Elektryczności Atmosferycznej i Meteorologii 1957, Prace Obs. Geof. Świder 16.

Warzecha S. (Ed.), 1961: Rocznik Elektryczności Atmosferycznej i Meteorologii 1958, Prace Obs. Geof. Świder 19.

Warzecha S. (Ed.), 1961: Rocznik Elektryczności Atmosferycznej i Meteorologii 1959, Prace Obs. Geof. Świder 20.

Warzecha S. (Ed.), 1962: Rocznik Elektryczności Atmosferycznej i Meteorologii 1960, Prace Obs. Geof. Świder 22.

Warzecha S. (Ed.), 1963: Rocznik Elektryczności Atmosferycznej i Meteorologii 1961, Prace Obs. Geof. Świder 25.

Warzecha S. (Ed.), 1964: Rocznik Elektrycznosci Atmosferycznej i Meteorologii 1962, Prace Obs. Geof. Świder 29.

Warzecha S. (Ed.), 1966: Rocznik Elektryczności Atmosferycznej i Meteorologii 1963, Prace Obs. Geof. Świder 33.

Warzecha S. (Ed.), 1967: Rocznik Elektryczności Atmosferycznej i Meteorologii 1964, Prace Obs. Geof. Świder 34.

Yearbooks including data from 1965-2005

Warzecha S. (Ed.), 1968: Rocznik Elektryczności Atmosferycznej i Meteorologii 1965, Prace Obs. Geof. Świder 38.

Warzecha S. (Ed.), 1968: Électricité atmosphérique et météorologie Observatoire Géophysique de St. Kalinowski à Świder 1966, Mater. Prace Zakł. Geofizyki 23.

Warzecha S. (Ed.), 1969: Électricité atmosphérique et météorologie Observatoire Géophysique de St. Kalinowski à Świder 1967, Mater. Prace Zakł. Geofizyki 28.

Warzecha S. (Ed.), 1970: Électricité atmosphérique et météorologie Observatoire Géophysique de St. Kalinowski à Świder 1968, Mater. Prace Zakł. Geofizyki 38.

Warzecha S. (Ed.), 1971: Électricité atmosphérique et météorologie Observatoire Géophysique de St. Kalinowski à Świder 1969, Mater. Prace Zakł. Geofizyki 44.

Warzecha S. (Ed.), 1972: Électricité atmosphérique et météorologie Observatoire Géophysique de St. Kalinowski à Świder

1970, Mater. Prace Zakł. Geofizyki 53.

Warzecha S. (Ed.), 1973: Électricité atmosphérique et météorologie Observatoire Géophysique de St. Kalinowski à Świder 1971, Mater. Prace Zakł. Geofizyki 63.

Warzecha S. (Ed.), 1974: Électricité atmosphérique et météorologie Observatoire Géophysique de St. Kalinowski à Świder 1972, Mater. Prace Zakł. Geofizyki 77.

Warzecha S. (Ed.), 1974: Électricité atmosphérique et météorologie Observatoire Géophysique de St. Kalinowski à Świder 1973, Mater. Prace Zakł. Geofizyki 80.

Warzecha S. (Ed.), 1976: Électricité atmosphérique et météorologie Observatoire Géophysique de St. Kalinowski à Świder 1974, Mater. Prace Zakł. Geofizyki 92.

Warzecha S. (Ed.), 1977: Électricité atmosphérique et météorologie Observatoire Géophysique de S. Kalinowski à Świder 1975, Publs. Inst. Geophys. Pol. Acad. Sci. 104 (D-2).

Warzecha S. (Ed.), 1978: Électricité atmosphérique et météorologie Observatoire Géophysique de S. Kalinowski à Świder 1976, Publs. Inst. Geophys. Pol. Acad. Sci. 121 (D-6).

Warzecha S. (Ed.), 1979: Électricité atmosphérique et météorologie Observatoire Géophysique de S. Kalinowski à Świder 1977, Publs. Inst. Geophys. Pol. Acad. Sci. 131 (D-8).

Warzecha S. (Ed.), 1980: Électricité atmosphérique et météorologie Observatoire Géophysique de S. Kalinowski à Świder 1978, Publs. Inst. Geophys. Pol. Acad. Sci. 140 (D-10).

Warzecha S. (Ed.), 1981: Électricité atmosphérique et météorologie Observatoire Géophysique de S. Kalinowski à Świder 1979, Publs. Inst. Geophys. Pol. Acad. Sci. 148 (D-12).

Warzecha S. (Ed.), 1982: Électricité atmosphérique et météorologie Observatoire Géophysique de S. Kalinowski à Świder 1980, Publs. Inst. Geophys. Pol. Acad. Sci. 151 (D-14).

Warzecha S. (Ed.), 1982: Électricité atmosphérique et météorologie Observatoire Géophysique de S. Kalinowski à Świder 1981, Publs. Inst. Geophys. Pol. Acad. Sci. 158 (D-16).

Warzecha S. (Ed.), 1983: Électricité atmosphérique et météorologie Observatoire Géophysique de S. Kalinowski à Świder 1982, Publs. Inst. Geophys. Pol. Acad. Sci. 168 (D-17).

Warzecha S. (Ed.), 1984: Électricité atmosphérique et météorologie Observatoire Géophysique de S. Kalinowski à Świder 1983, Publs. Inst. Geophys. Pol. Acad. Sci. 177 (D-19).

Warzecha S. (Ed.), 1985: Électricité atmosphérique et météorologie Observatoire Géophysique de S. Kalinowski à Świder 1984, Publs. Inst. Geophys. Pol. Acad. Sci. 190 (D-23).

Warzecha S. (Ed.), 1986: Électricité atmosphérique et météorologie Observatoire Géophysique de S. Kalinowski à Świder 1985, Publs. Inst. Geophys. Pol. Acad. Sci. 194 (D-24).

Warzecha S. (Ed.), 1987: Électricité atmosphérique et météorologie Observatoire Géophysique de S. Kalinowski à Świder 1986, Publs. Inst. Geophys. Pol. Acad. Sci. 209 (D-27).

Warzecha S. (Ed.), 1988: Électricité atmosphérique et météorologie Observatoire Géophysique de S. Kalinowski à Świder 1987, Publs. Inst. Geophys. Pol. Acad. Sci. 219 (D-29).

Warzecha S. (Ed.), 1989: Électricité atmosphérique et météorologie Observatoire Géophysique de S. Kalinowski à Świder

1988, Publs. Inst. Geophys. Pol. Acad. Sci. 229 (D-31).

Warzecha S. (Ed.), 1991: Électricité atmosphérique et météorologie Observatoire Géophysique de S. Kalinowski à Świder 1989, Publs. Inst. Geophys. Pol. Acad. Sci. 234 (D-34).

Warzecha S. (Ed.), 1991: Électricité atmosphérique et météorologie Observatoire Géophysique de S. Kalinowski à Świder 1990, Publs. Inst. Geophys. Pol. Acad. Sci. 247 (D-37).

Warzecha S. (Ed.), 1992: Électricité atmosphérique et météorologie Observatoire Géophysique de S. Kalinowski à Świder 1991, Publs. Inst. Geophys. Pol. Acad. Sci. 253 (D-39).

Warzecha S. (Ed.), 1993: Électricité atmosphérique et météorologie Observatoire Géophysique de S. Kalinowski à Świder 1992, Publs. Inst. Geophys. Pol. Acad. Sci. 264 (D-41).

Warzecha S. (Ed.), 1994: Électricité atmosphérique et météorologie Observatoire Géophysique de S. Kalinowski à Świder 1993, Publs. Inst. Geophys. Pol. Acad. Sci. 271 (D-43).

Warzecha S. (Ed.), 1995: Électricité atmosphérique et météorologie Observatoire Géophysique de S. Kalinowski à Świder 1994, Publs. Inst. Geophys. Pol. Acad. Sci. 280 (D-44).

Warzecha S. (Ed.), 1997: Électricité atmosphérique et météorologie Observatoire Géophysique de S. Kalinowski à Świder 1995, Publs. Inst. Geophys. Pol. Acad. Sci. 290 (D-47).

Kubicki M. (Ed.), 1998: Results of atmospheric electricity and meteorological observations. S.Kalinowski Geophysical Observatory at Świder – 1996, Publs. Inst. Geophys. Pol. Acad. Sci. 299 (D-49).

Kubicki M. (Ed.), 1999: Results of atmospheric electricity and meteorological observations. S.Kalinowski Geophysical Observatory at Świder – 1997, Publs. Inst. Geophys. Pol. Acad. Sci. 307 (D-51).

Kubicki M. (Ed.), 1999: Results of atmospheric electricity and meteorological observations. S.Kalinowski Geophysical Observatory at Świder – 1998, Publs. Inst. Geophys. Pol. Acad. Sci. 321 (D-52).

Kubicki M. (Ed.), 2000: Results of atmospheric electricity and meteorological observations. S.Kalinowski Geophysical Observatory at Świder – 1999, Publs. Inst. Geophys. Pol. Acad. Sci. 324 (D-54).

Kubicki M. (Ed.), 2001: Results of atmospheric electricity and meteorological observations. S.Kalinowski Geophysical Observatory at Świder – 2000, Publs. Inst. Geophys. Pol. Acad. Sci. 333 (D-56).

Kubicki M. (Ed.), 2002: Results of atmospheric electricity and meteorological observations. S.Kalinowski Geophysical Observatory at Świder – 2001, Publs. Inst. Geophys. Pol. Acad. Sci. 342 (D-58).

Kubicki M. (Ed.), 2003: Results of atmospheric electricity and meteorological observations. S.Kalinowski Geophysical Observatory at Świder – 2002, Publs. Inst. Geophys. Pol. Acad. Sci. 355 (D-61).

Kubicki M. (Ed.), 2004: Results of atmospheric electricity and meteorological observations. S.Kalinowski Geophysical Observatory at Świder – 2003, Publs. Inst. Geophys. Pol. Acad. Sci. 372 (D-65).

Kubicki M. (Ed.), 2005: Results of atmospheric electricity and meteorological observations. S.Kalinowski Geophysical Observatory at Świder – 2004, Publs. Inst. Geophys. Pol. Acad. Sci. 383 (D-68).

Kubicki M. (Ed.), 2006: Results of atmospheric electricity and meteorological observations. S.Kalinowski Geophysical Obser-

640 vatory at Świder, 2005, Publs. Inst. Geophys. Pol. Acad. Sci. 391 (D-71).

**Appendix B**

This Appendix contains the list of additional Tables: Table B1, Table B2 and Table B3.

*Author contributions.* AO digitised PG and CN data from Świder yearbooks and conceived the subject of the study. IZ and AO performed
the investigation, wrote and revised the paper. DK and JT revised and edited the paper. All authors discussed the results.

*Competing interests.* The authors declare that they have no competing interests.

*Acknowledgements.* The work is supported by Poland National Science Centre grant No 2021/41/B/ST10/04448 at the Institute of Geophysics, Polish Academy of Sciences.

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

**Table 4.** Mean values of PG in fair weather conditions by month for all and limited CN concentration values together with the standard error of the mean value corresponding to 6, 12, 18 UT.

| Hour | PG (V/m) | Jan | Feb | Mar | Apr | May | Jun | Jul | Aug | Sep | Oct | Nov | Dec |
|---|---|---|---|---|---|---|---|---|---|---|---|---|---|
| All | CN all | 349±5 | 361±5 | 320±4 | 239±3 | 205±2 | 187±2 | 191±2 | 200±2 | 230±3 | 257±4 | 290±5 | 339±6 |
| | CN<10000 | 323±11 | 283±9 | 307±10 | 229±7 | 202±4 | 184±4 | 190±3 | 201±4 | 214±6 | 247±8 | 255±9 | 309±11 |
| | CN<8000 | 321±15 | 274±13 | 300±14 | 230±8 | 202±6 | 187±5 | 192±4 | 198±5 | 212±8 | 235±9 | 254±12 | 303±14 |
| | CN<6000 | 274±19 | 266±20 | 270±14 | 219±10 | 195±8 | 187±7 | 187±5 | 197±6 | 216±10 | 229±14 | 232±15 | 294±22 |
| 6 UT | CN all | 294±9 | 286±9 | 298±8 | 248±5 | 227±4 | 207±4 | 216±3 | 220±4 | 226±6 | 213±6 | 256±10 | 295±11 |
| | CN<10000 | 273±15 | 236±14 | 283±16 | 233±10 | 211±6 | 195±5 | 213±5 | 212±6 | 197±10 | 208±12 | 232±17 | 279±15 |
| | CN<8000 | 258±22 | 215±16 | 277±20 | 240±13 | 215±9 | 195±7 | 218±7 | 206±6 | 188±15 | 208±14 | 243±21 | 287±21 |
| | CN<6000 | 209±20 | 200±20 | 228±20 | 236±21 | 211±11 | 179±12 | 209±10 | 211±10 | 212±30 | 210±18 | 204±19 | 256±29 |
| 12 UT | CN all | 372±8 | 378±8 | 298±6 | 202±4 | 170±4 | 165±4 | 171±3 | 170±3 | 214±4 | 259±5 | 298±7 | 356±9 |
| | CN<10000 | 367±21 | 310±14 | 299±13 | 208±7 | 169±7 | 155±7 | 168±4 | 174±5 | 222±7 | 240±9 | 262±12 | 343±24 |
| | CN<8000 | 380±29 | 295±18 | 311±17 | 210±9 | 173±8 | 158±10 | 167±5 | 175±6 | 216±9 | 233±10 | 269±17 | 332±37 |
| | CN<6000 | 334±29 | 333±39 | 303±13 | 206±10 | 166±11 | 184±13 | 169±6 | 175±7 | 221±11 | 233±17 | 233±32 | 357±36 |
| 18 UT | CN all | 364±10 | 395±10 | 355±7 | 253±5 | 202±3 | 178±3 | 178±3 | 201±3 | 247±6 | 287±8 | 301±10 | 351±11 |
| | CN<10000 | 339±17 | 303±17 | 359±24 | 250±15 | 205±6 | 181±5 | 179±4 | 211±7 | 221±12 | 304±18 | 271±19 | 316±16 |
| | CN<8000 | 338±22 | 309±24 | 333±48 | 261±22 | 196±10 | 188±6 | 186±5 | 216±11 | 232±23 | 289±25 | 254±19 | 303±22 |
| | CN<6000 | 344±38 | 293±35 | 305±47 | 240±35 | 190±22 | 194±11 | 185±7 | 229±21 | 203±26 | 271±73 | 268±27 | 331±55 |

**Table 5.** Model of ion conductivity with CN concentrations dominated by soot: concentrations of the CN components, the ion-aerosol loss term $\beta N$, ion conductivity, and conductivity ratios at different concentration ratio of water soluble particles and soot components. A constant concentration of insoluble aerosol at 500 cm$^{-3}$ is added to the total CN concentration.

| Soot (%) | WatSol (%) | $N_{Soot}$ (cm$^{-3}$) | $N_{WatSol}$ (cm$^{-3}$) | CN (cm$^{-3}$) | $\beta N$ (10$^{-2}$ s$^{-1}$) | $\beta_{eff}$ (10$^{-6}$ cm$^3$s$^{-1}$) | Conductivity (10$^{-15}$ S/m) | s | |
|---|---|---|---|---|---|---|---|---|---|
| 100 | 0 | 26000 | 0 | 26500 | 2.85 | 1.07 | 4.20 | 1.00 | |
| 90 | 10 | 23400 | 600 | 24500 | 2.77 | 1.13 | 4.32 | 1.03 | |
| 80 | 20 | 20800 | 1200 | 22500 | 2.70 | 1.20 | 4.43 | 1.05 | |
| 70 | 30 | 18200 | 1800 | 20500 | 2.62 | 1.28 | 4.56 | 1.08 | W mean |
| 60 | 40 | 15600 | 2400 | 18500 | 2.55 | 1.38 | 4.69 | 1.11 | W median |
| 50 | 50 | 13000 | 3000 | 16500 | 2.48 | 1.50 | 4.82 | 1.15 | |
| 40 | 60 | 10400 | 3600 | 14500 | 2.40 | 1.66 | 4.97 | 1.18 | S mean |
| 30 | 70 | 7800 | 4200 | 12500 | 2.33 | 1.86 | 5.13 | 1.22 | S median |
| 20 | 80 | 5200 | 4800 | 10500 | 2.26 | 2.15 | 5.29 | 1.26 | |
| 10 | 90 | 2600 | 5400 | 8500 | 2.18 | 2.57 | 5.47 | 1.30 | |
| 0 | 100 | 0 | 6000 | 6500 | 2.11 | 3.24 | 5.66 | 1.35 | |

**Table 6.** Model of ion conductivity with CN concentrations dominated by water soluble particles: concentrations of the CN components, the ion-aerosol loss term $\beta N$, ion conductivity, and conductivity ratios at different concentration ratio of water soluble particles and soot components. A constant concentration of insoluble aerosol at 200 cm$^{-3}$ is added to the total CN concentration.

| Soot (%) | WatSol (%) | $N_{Soot}$ (cm$^{-3}$) | $N_{WatSol}$ (cm$^{-3}$) | CN (cm$^{-3}$) | $\beta N$ (10$^{-2}$ s$^{-1}$) | $\beta_{eff}$ (10$^{-6}$ cm$^3$s$^{-1}$) | Conductivity (10$^{-15}$ S/m) | s | |
|---|---|---|---|---|---|---|---|---|---|
| 0 | 100 | 0 | 28000 | 28200 | 3.57 | 1.37 | 3.10 | 1.00 | |
| 10 | 90 | 600 | 25200 | 26000 | 3.57 | 1.37 | 3.36 | 1.08 | |
| 20 | 80 | 1200 | 22400 | 23800 | 3.27 | 1.37 | 3.66 | 1.18 | |
| 30 | 70 | 1800 | 19600 | 21600 | 2.97 | 1.38 | 4.03 | 1.30 | W mean |
| 40 | 60 | 2400 | 16800 | 19400 | 2.67 | 1.38 | 4.47 | 1.44 | W median |
| 50 | 50 | 3000 | 14000 | 17200 | 2.38 | 1.38 | 5.02 | 1.62 | |
| 60 | 40 | 3600 | 11200 | 15000 | 2.08 | 1.38 | 5.73 | 1.85 | S mean |
| 70 | 30 | 4200 | 8400 | 12800 | 1.78 | 1.39 | 6.67 | 2.15 | S median |
| 80 | 20 | 4800 | 5600 | 10600 | 1.48 | 1.40 | 7.97 | 2.57 | |
| 90 | 10 | 5400 | 2800 | 8400 | 1.19 | 1.41 | 9.88 | 3.18 | |
| 100 | 0 | 6000 | 0 | 6200 | 0.89 | 1.44 | 13.0 | 4.17 | |

**Table 7.** Mean values of fair weather positive conductivity (PC) by season for all and limited CN concentration values together with the one standard error of the mean.

|  | Winter | Spring | Summer | Autumn | All year |
|---|---|---|---|---|---|
| All values | $2.24 \pm 0.01$ | $3.20 \pm 0.01$ | $3.98 \pm 0.01$ | $3.09 \pm 0.01$ | $3.28 \pm 0.01$ |
| CN all | $2.16 \pm 0.02$ | $3.14 \pm 0.03$ | $4.15 \pm 0.02$ | $2.98 \pm 0.02$ | $3.27 \pm 0.01$ |
| CN<10000 | $2.40 \pm 0.06$ | $3.69 \pm 0.09$ | $4.37 \pm 0.04$ | $3.35 \pm 0.06$ | $3.79 \pm 0.03$ |
| CN<8000 | $2.38 \pm 0.07$ | $3.72 \pm 0.10$ | $4.40 \pm 0.05$ | $3.50 \pm 0.07$ | $3.85 \pm 0.04$ |
| CN<6000 | $2.52 \pm 0.11$ | $3.71 \pm 0.15$ | $4.48 \pm 0.06$ | $3.36 \pm 0.09$ | $3.91 \pm 0.06$ |

**Table 8.** Mean value and standard error of air positive conductivity in fair weather conditions (mean of 3 hours: 6, 12, 18 UT, and at each hour) by month for all and limited CN concentrations.

| Hour | PC (fS/m) | Jan | Feb | Mar | Apr | May | Jun | Jul | Aug | Sep | Oct | Nov | Dec |
|---|---|---|---|---|---|---|---|---|---|---|---|---|---|
| All | CN all | 2.28±0.04 | 1.98±0.03 | 2.30±0.03 | 2.96±0.05 | 3.95±0.04 | 4.28±0.04 | 4.32±0.04 | 3.94±0.04 | 3.20±0.04 | 2.88±0.04 | 2.69±0.05 | 2.24±0.04 |
| | CN<10000 | 2.60±0.11 | 2.34±0.10 | 2.47±0.09 | 3.23±0.10 | 4.24±0.10 | 4.36±0.07 | 4.49±0.06 | 4.27±0.06 | 3.59±0.09 | 3.31±0.09 | 2.88±0.09 | 2.25±0.09 |
| | CN<8000 | 2.60±0.13 | 2.29±0.13 | 2.61±0.12 | 3.34±0.14 | 4.29±0.14 | 4.31±0.09 | 4.30±0.07 | 4.30±0.08 | 3.76±0.10 | 3.48±0.10 | 3.02±0.14 | 2.25±0.12 |
| | CN<6000 | 2.90±0.20 | 2.27±0.20 | 2.61±0.14 | 3.23±0.12 | 4.63±0.27 | 4.47±0.12 | 4.63±0.10 | 4.34±0.11 | 3.48±0.13 | 3.46±0.13 | 3.06±0.18 | 2.41±0.18 |
| 6 UT | CN all | 2.37±0.08 | 1.94±0.07 | 2.30±0.06 | 3.09±0.08 | 4.03±0.08 | 4.10±0.06 | 4.28±0.06 | 4.06±0.05 | 3.40±0.07 | 2.81±0.07 | 2.67±0.11 | 2.30±0.10 |
| | CN<10000 | 2.61±0.17 | 2.51±0.19 | 2.57±0.14 | 3.12±0.13 | 4.38±0.17 | 4.20±0.11 | 4.41±0.09 | 4.29±0.09 | 3.67±0.16 | 3.38±0.18 | 2.96±0.18 | 2.37±0.18 |
| | CN<8000 | 2.72±0.22 | 2.62±0.27 | 2.68±0.18 | 3.18±0.19 | 4.40±0.25 | 4.06±0.13 | 4.45±0.11 | 4.33±0.13 | 4.06±0.25 | 3.16±0.14 | 3.16±0.23 | 2.36±0.19 |
| | CN<6000 | 2.99±0.29 | 3.03±0.39 | 2.80±0.22 | 3.20±0.29 | 4.77±0.45 | 4.38±0.20 | 3.32±0.14 | 4.37±0.18 | 3.32±0.24 | 3.52±0.23 | 3.18±0.26 | 2.45±0.24 |
| 12 UT | CN all | 2.48±0.06 | 2.37±0.06 | 2.89±0.06 | 3.44±0.10 | 3.88±0.09 | 4.06±0.09 | 4.13±0.09 | 4.21±0.08 | 3.72±0.07 | 3.42±0.06 | 2.99±0.07 | 2.48±0.07 |
| | CN<10000 | 2.65±0.25 | 2.43±0.17 | 2.73±0.14 | 3.37±0.15 | 4.22±0.18 | 4.32±0.15 | 4.74±0.15 | 4.59±0.12 | 3.51±0.11 | 3.48±0.12 | 2.94±0.14 | 2.29±0.17 |
| | CN<8000 | 2.50±0.27 | 2.41±0.22 | 2.79±0.21 | 3.38±0.17 | 4.08±0.19 | 4.36±0.16 | 4.78±0.16 | 4.53±0.14 | 3.65±0.13 | 3.81±0.16 | 3.03±0.25 | 2.31±0.33 |
| | CN<6000 | 2.83±0.53 | 2.11±0.38 | 2.53±0.19 | 3.27±0.12 | 4.30±0.22 | 4.44±0.24 | 4.85±0.16 | 4.50±0.17 | 3.65±0.15 | 3.54±0.18 | 2.73±0.20 | 2.51±0.47 |
| 18 UT | CN all | 1.99±0.07 | 1.62±0.05 | 1.78±0.05 | 2.55±0.09 | 3.92±0.08 | 4.55±0.07 | 4.46±0.07 | 3.65±0.06 | 2.62±0.06 | 2.34±0.06 | 2.35±0.08 | 1.98±0.07 |
| | CN<10000 | 2.53±0.17 | 2.06±0.12 | 1.89±0.14 | 3.23±0.26 | 4.04±0.13 | 4.52±0.11 | 4.44±0.10 | 3.95±0.11 | 3.61±0.21 | 2.92±0.23 | 2.72±0.18 | 2.13±0.14 |
| | CN<8000 | 2.57±0.20 | 1.86±0.14 | 2.04±0.22 | 3.65±0.55 | 4.25±0.19 | 4.51±0.14 | 4.52±0.12 | 3.93±0.13 | 3.62±0.33 | 3.12±0.20 | 2.83±0.24 | 2.09±0.16 |
| | CN<6000 | 2.70±0.15 | 1.69±0.13 | 2.23±0.35 | 3.09±0.45 | 4.64±0.51 | 4.57±0.18 | 4.66±0.20 | 3.79±0.23 | 3.07±0.36 | 2.93±0.40 | 3.12±0.42 | 2.14±0.20 |

**Table B1.** Number of hourly values of PG at fair weather hours divided into months for January 1965 – December 2005. Last column indicates the number of hours without PG measurements.

| Season | Winter | | | Spring | | | Summer | | | Autumn | | | total | missing |
|---|---|---|---|---|---|---|---|---|---|---|---|---|---|---|
| Y\M | Dec | Jan | Feb | Mar | Apr | May | Jun | Jul | Aug | Sep | Oct | Nov | | |
| 1965 | 105 | 211 | 148 | 315 | 205 | 275 | 286 | 342 | 332 | 440 | 270 | 58 | 2987 | 59 |
| 1966 | 125 | 85 | 193 | 226 | 340 | 289 | 462 | 391 | 476 | 407 | 324 | 99 | 3417 | 91 |
| 1967 | 104 | 242 | 268 | 256 | 288 | 383 | 370 | 443 | 445 | 395 | 385 | 222 | 3801 | 152 |
| 1968 | 138 | 100 | 146 | 340 | 427 | 342 | 384 | 339 | 529 | 364 | 244 | 116 | 3469 | 109 |
| 1969 | 181 | 257 | 177 | 308 | 346 | 361 | 401 | 374 | 354 | 394 | 235 | 110 | 3498 | 111 |
| 1970 | 31 | 199 | 197 | 152 | 224 | 271 | 410 | 297 | 324 | 203 | 91 | 113 | 2512 | 24 |
| 1971 | 33 | 275 | 106 | 241 | 309 | 404 | 300 | 512 | 526 | 221 | 207 | 143 | 3277 | 99 |
| 1972 | 300 | 214 | 276 | 279 | 200 | 320 | 347 | 433 | 237 | 149 | 251 | 127 | 3133 | 68 |
| 1973 | 174 | 184 | 46 | 346 | 287 | 311 | 326 | 273 | 552 | 305 | 209 | 181 | 3194 | 107 |
| 1974 | 104 | 236 | 217 | 501 | 347 | 239 | 261 | 201 | 432 | 376 | 150 | 207 | 3271 | 142 |
| 1975 | 137 | 195 | 204 | 256 | 191 | 339 | 309 | 373 | 473 | 430 | 167 | 182 | 3256 | 103 |
| 1976 | 94 | 124 | 321 | 178 | 281 | 321 | 318 | 397 | 382 | 289 | 199 | 140 | 3044 | 64 |
| 1977 | 96 | 252 | 98 | 291 | 227 | 299 | 390 | 304 | 287 | 281 | 189 | 146 | 2860 | 68 |
| 1978 | 129 | 237 | 120 | 230 | 271 | 374 | 351 | 268 | 143 | 67 | 69 | 41 | 2300 | 14 |
| 1979 | 128 | 45 | 237 | 150 | 233 | 401 | 290 | 119 | 171 | 156 | 346 | 63 | 2339 | 110 |
| 1980 | 95 | 126 | 86 | 242 | 190 | 247 | 167 | 90 | 181 | 143 | 179 | 99 | 1845 | 47 |
| 1981 | 109 | 102 | 165 | 225 | 228 | 309 | 229 | 243 | 277 | 185 | 132 | 35 | 2239 | 87 |
| 1982 | 21 | 129 | 158 | 316 | 136 | 383 | 230 | 373 | 408 | 361 | 239 | 200 | 2954 | 98 |
| 1983 | 126 | 60 | 131 | 180 | 264 | 271 | 359 | 356 | 461 | 253 | 150 | 89 | 2700 | 67 |
| 1984 | 56 | 63 | 259 | 281 | 346 | 279 | 130 | 234 | 441 | 149 | 275 | 156 | 2669 | 87 |
| 1985 | 70 | 164 | 153 | 124 | 326 | 399 | 205 | 358 | 373 | 176 | 89 | 76 | 2513 | 42 |
| 1986 | 108 | 95 | 303 | 295 | 323 | 421 | 351 | 311 | 334 | 146 | 259 | 136 | 3082 | 85 |
| 1987 | 69 | 212 | 147 | 292 | 208 | 284 | 182 | 343 | 288 | 288 | 431 | 132 | 2876 | 80 |
| 1988 | 84 | 174 | 168 | 187 | 409 | 328 | 286 | 419 | 370 | 203 | 403 | 171 | 3202 | 42 |
| 1989 | 156 | 75 | 152 | 262 | 256 | 425 | 201 | 403 | 296 | 346 | 145 | 64 | 2781 | 54 |
| 1990 | 106 | 119 | 208 | 264 | 318 | 355 | 328 | 282 | 392 | 174 | 320 | 25 | 2891 | 82 |
| 1991 | 128 | 197 | 123 | 165 | 309 | 311 | 281 | 405 | 333 | 422 | 291 | 183 | 3148 | 73 |
| 1992 | 224 | 166 | 100 | 275 | 243 | 391 | 470 | 485 | 543 | 459 | 195 | 174 | 3725 | 74 |
| 1993 | 119 | 326 | 164 | 366 | 401 | 517 | 411 | 398 | 427 | 252 | 274 | 157 | 3812 | 56 |
| 1994 | 104 | 178 | 225 | 169 | 373 | 407 | 380 | 614 | 378 | 245 | 248 | 231 | 3552 | 81 |
| 1995 | 184 | 156 | 244 | 174 | 317 | 352 | 341 | 553 | 496 | 188 | 185 | 161 | 3351 | 42 |
| 1996 | 304 | 302 | 220 | 272 | 456 | 279 | 398 | 329 | 447 | 186 | 247 | 227 | 3667 | 152 |
| 1997 | 155 | 191 | 221 | 370 | 331 | 373 | 334 | 285 | 536 | 319 | 219 | 180 | 3514 | 65 |
| 1998 | 172 | 281 | 222 | 379 | 290 | 405 | 297 | 312 | 274 | 342 | 265 | 146 | 3385 | 243 |
| 1999 | 202 | 220 | 110 | 360 | 303 | 442 | 271 | 471 | 434 | 478 | 259 | 214 | 3764 | 79 |
| 2000 | 137 | 123 | 221 | 200 | 444 | 543 | 464 | 265 | 426 | 425 | 595 | 265 | 4108 | 100 |
| 2001 | 143 | 156 | 245 | 310 | 255 | 479 | 335 | 316 | 511 | 232 | 249 | 171 | 3402 | 90 |
| 2002 | 273 | 158 | 336 | 370 | 348 | 491 | 398 | 441 | 525 | 377 | 180 | 289 | 4186 | 39 |
| 2003 | 298 | 216 | 277 | 314 | 352 | 482 | 382 | 373 | 501 | 455 | 186 | 166 | 4002 | 50 |
| 2004 | 180 | 153 | 162 | 221 | 431 | 319 | 333 | 415 | 463 | 453 | 292 | 187 | 3609 | 25 |
| 2005 | 41 | 219 | 233 | 356 | 428 | 453 | 390 | 430 | 536 | 552 | 455 | 206 | 4299 | 42 |
| total | 5543 | 7217 | 7787 | 11038 | 12461 | 14874 | 13358 | 14570 | 16314 | 12286 | 10098 | 6088 | 131634 | 3303 |
| | | 20547 | | | 38373 | | | 44242 | | | 28472 | | | |

**Table B2.** Number of fair weather days (24h) divided into months for the period January 1965 – December 2005.

| Season Y\M | Winter | | | Spring | | | Summer | | | Autumn | | | total |
| --- | --- | --- | --- | --- | --- | --- | --- | --- | --- | --- | --- | --- | --- |
| | Dec | Jan | Feb | Mar | Apr | May | Jun | Jul | Aug | Sep | Oct | Nov | |
| 1965 | – | 2 | – | 3 | – | 4 | 3 | 5 | 6 | 10 | 2 | – | 35 |
| 1966 | – | – | 3 | 1 | 5 | 5 | 11 | 1 | 11 | 4 | 2 | – | 43 |
| 1967 | 1 | 4 | 3 | 1 | 3 | 10 | 3 | 6 | 2 | 7 | 3 | 3 | 46 |
| 1968 | 1 | – | 1 | 3 | 10 | – | 6 | 3 | 7 | 4 | 2 | 2 | 39 |
| 1969 | 3 | 5 | 1 | 4 | 6 | 5 | 3 | 7 | 5 | 8 | 1 | – | 48 |
| 1970 | – | 2 | 1 | 1 | 2 | 4 | 6 | 1 | 5 | 2 | – | – | 24 |
| 1971 | – | 2 | – | 1 | 5 | 4 | 3 | 12 | 11 | 3 | 2 | – | 43 |
| 1972 | 4 | 4 | 6 | 6 | 2 | 3 | 5 | 9 | 2 | 1 | 1 | – | 43 |
| 1973 | 1 | 3 | – | 3 | 2 | 4 | 5 | 1 | 11 | 2 | 1 | 1 | 34 |
| 1974 | – | 4 | 3 | 11 | 4 | 2 | 1 | – | 10 | 6 | – | 2 | 43 |
| 1975 | – | 3 | 1 | 3 | 1 | 2 | 3 | 7 | 12 | 6 | 4 | 1 | 43 |
| 1976 | – | 2 | 6 | 1 | 1 | 5 | 7 | 7 | 5 | 4 | 1 | – | 39 |
| 1977 | 2 | 2 | – | 2 | 1 | 2 | 5 | 4 | 4 | 2 | 1 | 1 | 26 |
| 1978 | – | 5 | 1 | 2 | 3 | 7 | 5 | 1 | 1 | – | – | – | 25 |
| 1979 | – | – | 3 | – | 3 | 3 | 2 | – | 2 | – | 3 | – | 16 |
| 1980 | 1 | – | – | – | 2 | – | – | – | – | – | – | – | 3 |
| 1981 | – | – | 1 | 1 | 3 | 2 | 1 | 1 | 3 | – | – | – | 12 |
| 1982 | – | – | – | 4 | – | 10 | 4 | 2 | 6 | 2 | 3 | 2 | 33 |
| 1983 | – | – | – | 3 | 2 | 1 | 3 | 2 | 7 | 3 | – | – | 21 |
| 1984 | – | – | 5 | 3 | 3 | 1 | – | 3 | 5 | – | 1 | 2 | 23 |
| 1985 | – | 2 | 1 | – | 3 | 7 | 1 | – | 5 | – | – | – | 19 |
| 1986 | 1 | – | 6 | 4 | 5 | 6 | 2 | 3 | 3 | 1 | 2 | – | 33 |
| 1987 | – | 1 | 1 | 6 | 1 | 3 | 1 | 4 | 4 | 2 | 7 | – | 30 |
| 1988 | 1 | 2 | 1 | – | 7 | 3 | 2 | 5 | 1 | 1 | 6 | 3 | 32 |
| 1989 | 1 | – | – | 2 | 3 | 7 | 1 | 7 | 2 | 5 | – | – | 28 |
| 1990 | – | – | – | 4 | 6 | 6 | 1 | 2 | 6 | – | 2 | – | 27 |
| 1991 | – | 1 | – | 1 | 3 | 3 | 1 | 6 | 2 | 4 | 3 | – | 24 |
| 1992 | 2 | – | – | 3 | 1 | 5 | 4 | 9 | 9 | 10 | 1 | – | 44 |
| 1993 | – | 6 | – | 5 | 8 | 7 | 7 | 4 | 6 | 2 | 3 | 1 | 49 |
| 1994 | – | 2 | 1 | – | 2 | 3 | 7 | 16 | 2 | – | 3 | 4 | 40 |
| 1995 | 1 | 1 | – | – | 4 | 3 | 3 | 10 | 7 | – | – | – | 29 |
| 1996 | 2 | 4 | – | 1 | 6 | – | 9 | – | 5 | 1 | 2 | 1 | 31 |
| 1997 | 1 | – | 1 | 3 | 2 | 6 | 3 | – | 13 | 3 | 1 | 3 | 36 |
| 1998 | 1 | 2 | – | 5 | 1 | 5 | 4 | 3 | 3 | 4 | 2 | 1 | 31 |
| 1999 | 1 | 1 | – | 6 | 4 | 6 | 3 | 9 | 2 | 10 | 2 | 2 | 46 |
| 2000 | 1 | – | 1 | – | 6 | 12 | 8 | 1 | 5 | 6 | 16 | 1 | 57 |
| 2001 | – | – | 1 | – | 3 | 6 | 1 | – | 6 | 1 | 1 | – | 19 |
| 2002 | – | – | 3 | 6 | 2 | 6 | 5 | 8 | 14 | 5 | – | 3 | 52 |
| 2003 | 4 | 1 | 4 | 6 | 4 | 8 | 6 | 3 | 5 | 6 | – | – | 47 |
| 2004 | 2 | – | 3 | 1 | 10 | 2 | 1 | 3 | 3 | 5 | 4 | – | 34 |
| 2005 | – | 2 | 4 | 5 | 4 | 9 | – | 5 | 10 | 15 | 11 | 2 | 67 |
| total | 31 | 63 | 62 | 111 | 143 | 187 | 146 | 170 | 228 | 145 | 93 | 35 | 1414 |
| | | 156 | | | 441 | | | 544 | | | 273 | | |

**Table B3.** Number of PG values measured in FW conditions by month for all and limited CN concentrations corresponding to 6, 12, 18 UT.

| Hour | PG length | Jan | Feb | Mar | Apr | May | Jun | Jul | Aug | Sep | Oct | Nov | Dec |
|---|---|---|---|---|---|---|---|---|---|---|---|---|---|
| 6UT | all CN | 239 | 269 | 418 | 545 | 737 | 653 | 702 | 754 | 475 | 347 | 190 | 190 |
| | PG with all CN | 236 | 258 | 397 | 526 | 709 | 628 | 670 | 723 | 449 | 326 | 178 | 183 |
| | CN<10000 | 52 | 45 | 6 3 | 102 | 240 | 217 | 242 | 218 | 88 | 55 | 50 | 55 |
| | CN<8000 | 27 | 25 | 37 | 52 | 139 | 118 | 138 | 123 | 36 | 34 | 3 5 | 34 |
| | CN<6000 | 16 | 13 | 22 | 21 | 56 | 45 | 58 | 55 | 9 | 13 | 18 | 18 |
| 12 UT | all CN | 363 | 388 | 467 | 382 | 409 | 329 | 377 | 523 | 481 | 508 | 336 | 301 |
| | PG with all CN | 356 | 377 | 454 | 374 | 395 | 322 | 367 | 514 | 476 | 497 | 330 | 298 |
| | CN<10000 | 41 | 47 | 55 | 76 | 79 | 74 | 137 | 160 | 109 | 94 | 56 | 35 |
| | CN<8000 | 20 | 25 | 31 | 63 | 51 | 44 | 108 | 110 | 72 | 52 | 25 | 16 |
| | CN<6000 | 8 | 7 | 19 | 42 | 27 | 25 | 73 | 72 | 36 | 17 | 9 | 8 |
| 18 UT | all CN | 322 | 393 | 544 | 621 | 718 | 676 | 713 | 772 | 599 | 467 | 293 | 255 |
| | PG with all CN | 318 | 377 | 526 | 592 | 682 | 638 | 671 | 722 | 564 | 418 | 277 | 246 |
| | CN<10000 | 47 | 44 | 36 | 54 | 153 | 222 | 270 | 166 | 65 | 48 | 43 | 55 |
| | CN<8000 | 29 | 26 | 14 | 22 | 70 | 123 | 151 | 79 | 25 | 18 | 25 | 28 |
| | CN<6000 | 8 | 14 | 8 | 8 | 19 | 49 | 65 | 23 | 12 | 4 | 13 | 5 |

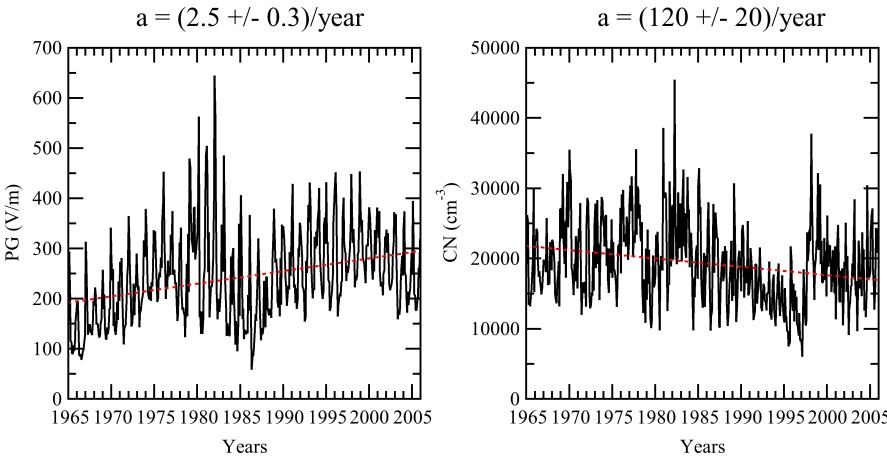

**Figure 1.** Time series of monthly averaged values of fair weather PG and CN values over January 1965 – December 2005. The dashed red lines indicate positive trend for PG (2.5±0.3 V/m per year) and negative trend for CN (–120±20 cm$^{-3}$).

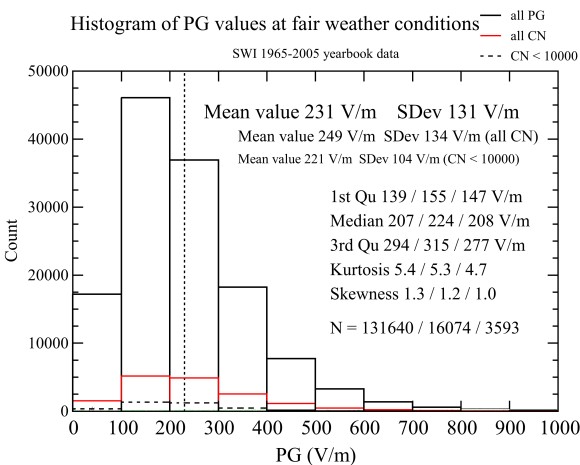

**Figure 2.** Histogram of the PG values measured during fair weather conditions in January 1965 – December 2005: black colour – all PG values, red colour – PG values recorded simultaneously with CN concentrations, black dashed line – the PG values at CN concentrations below 10000 cm$^{-3}$. The vertical black dashed line represents mean PG value (230 V/m). The other statistical parameters separately for the three PG groups are given in the legend, including the number of data values, N.

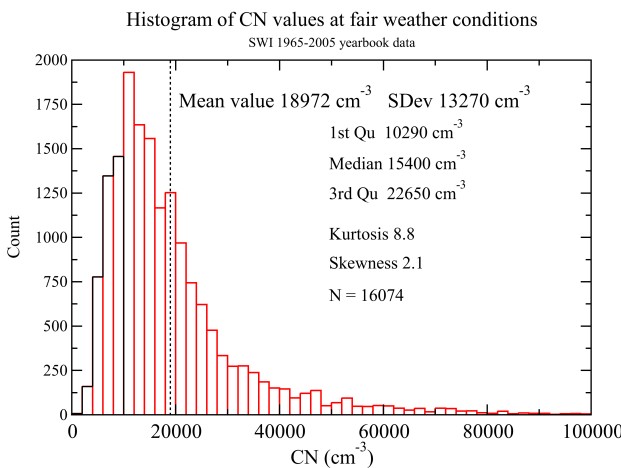

**Figure 3.** Histogram of hourly averaged CN concentrations at fair weather conditions in January 1965 – December 2005: CN below 10000 $\text{cm}^{-3}$ are framed in green, red colour – all CN. The black dashed line represents mean CN concentration value of 18980 $\text{cm}^{-3}$.

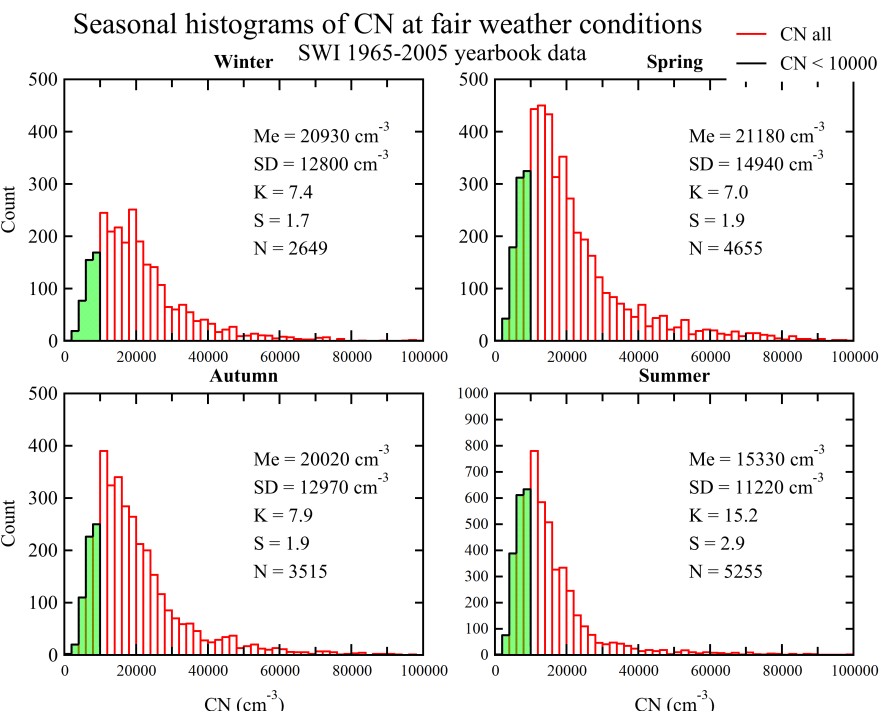

**Figure 4.** Seasonal histograms of the hourly averaged (6, 12, 18 UT) CN concentration values measured during fair weather conditions for the period January 1965 – December 2005: red colour – all CN concentration values, black colour – CN concentration values below 10000 cm$^{-3}$. Note that Y–axis ranges for summer (0-1000 counts) is twice compared with the other seasons (0-500 counts).

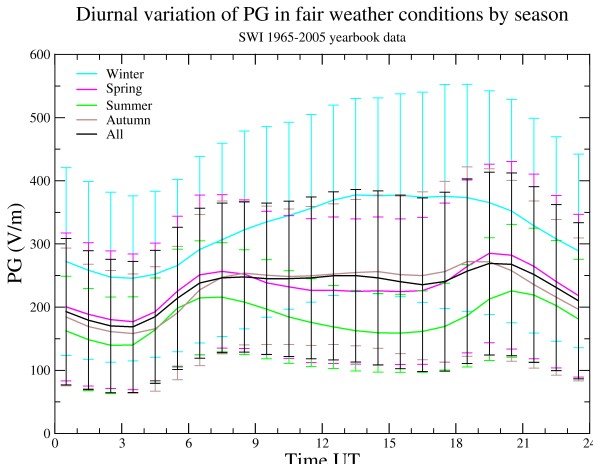

**Figure 5.** Diurnal variation of PG measured during fair weather conditions by seasons and for the whole year in January 1965 – December 2005. The error bars represent ±1 standard deviation.

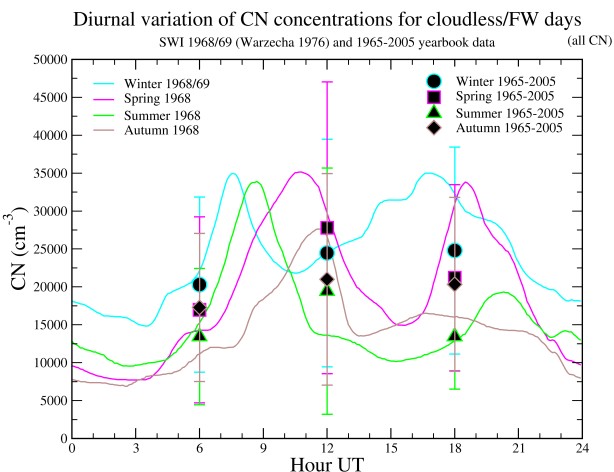

**Figure 6.** Diurnal variation of CN concentration values for cloudless days in spring 1968 – winter 1968/1969 and mean CN concentrations measured for 6, 12, 18 UT in 1965–2005.

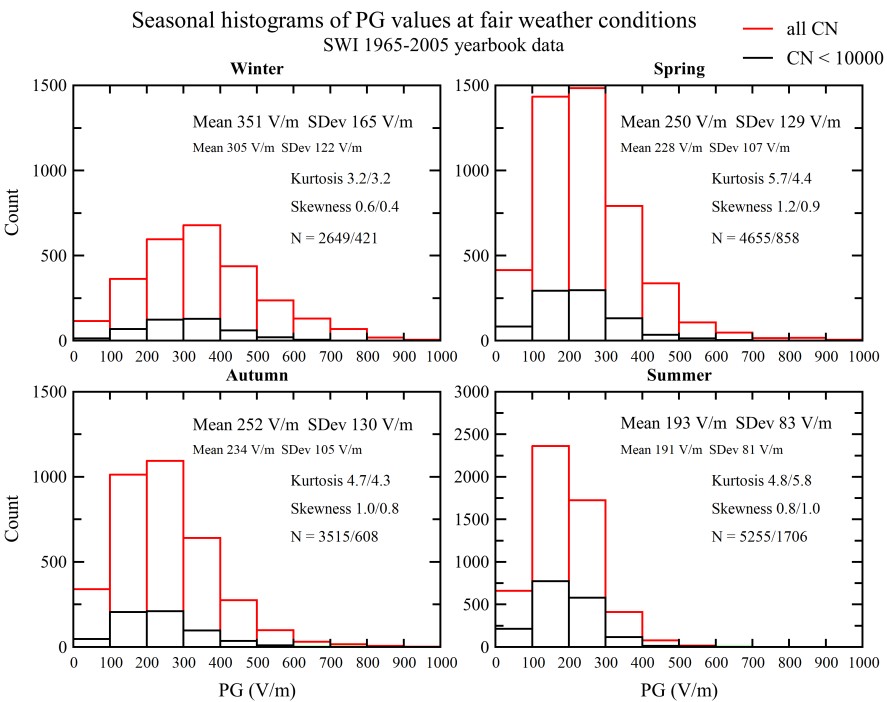

**Figure 7.** Seasonal histograms of the hourly averaged (6, 12, 18 UT) PG values measured during fair weather conditions in January 1965 – December 2005: red colour–PG values measured simultaneously with all CN concentration values (6, 12, 18 UT), black colour–PG values measured simultaneously with CN concentrations below 10000 cm$^{-3}$. Note that the Y–axis ranges for summer (0-3000 counts) is twice compared with the other seasons (0-1500 counts).

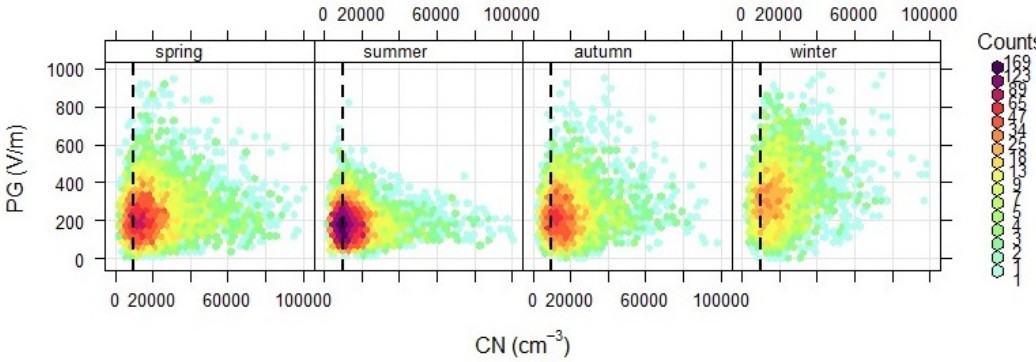

**Figure 8.** Seasonal dependence of hourly averaged values of fair weather PG and CN concentration values in January 1965 – December 2005. Horizontal dashed line indicates the limit of CN concentration at $10000\ \mathrm{cm}^{-3}$.

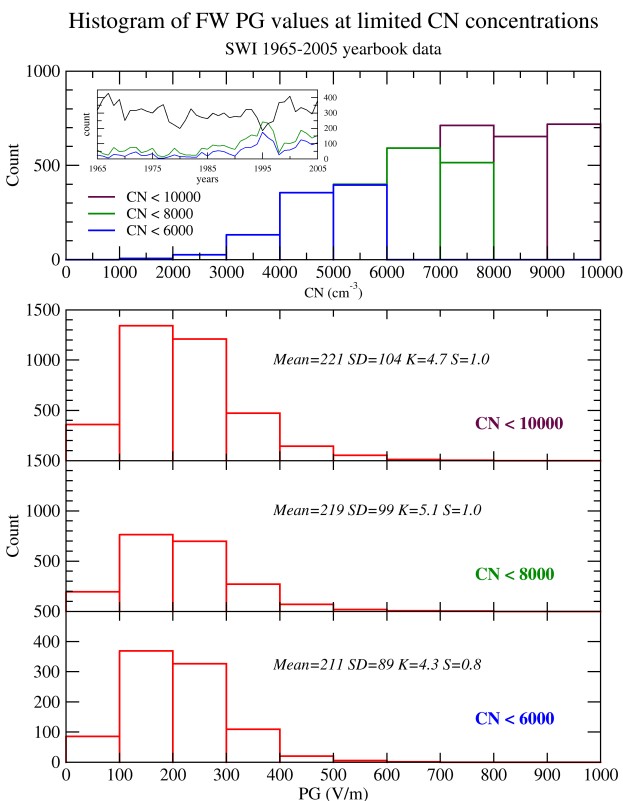

**Figure 9.** Histograms of the limited CN concentration values measured during fair weather conditions divided into CN range: below 10000 cm$^{-3}$ (purple), 8000 cm$^{-3}$ (green), and 6000 cm$^{-3}$ (blue), Bottom panels: histograms of hourly averaged PG values measured at limited CN concentration.

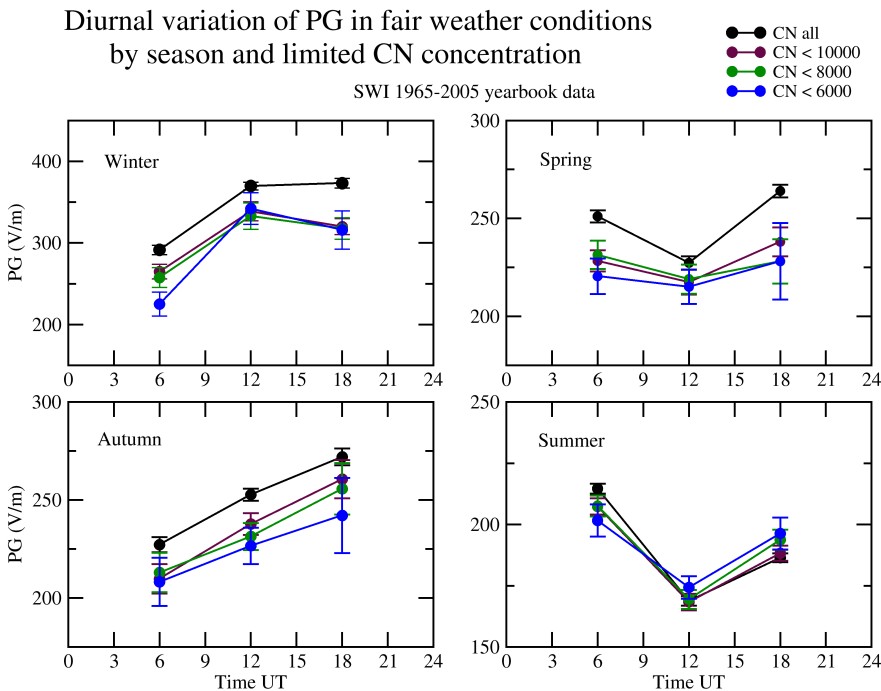

**Figure 10.** Diurnal variation of PG values measured during fair weather conditions at 6, 12, 18 UT, divided into seasons, for the period January 1965 – December 2005 for all CN concentration values (black), and CN concentration values below: 10000 $cm^{-3}$, 8000 $cm^{-3}$, 6000 $cm^{-3}$. Error bars indicate one standard error of the mean.

**Figure 11.** Annual variation of potential gradient in fair weather conditions for all and limited CN concentration values for all hours (upper chart) and separately for each hour (6, 12, 18 UT). Error bars indicate one standard error of the mean.

**Figure 12.** Annual variation of air positive conductivity, PC, in fair weather conditions for all and limited CN concentration values for all hours (upper chart) and separately for each hour (6, 12, 18 UT). Error bars indicate one standard error.

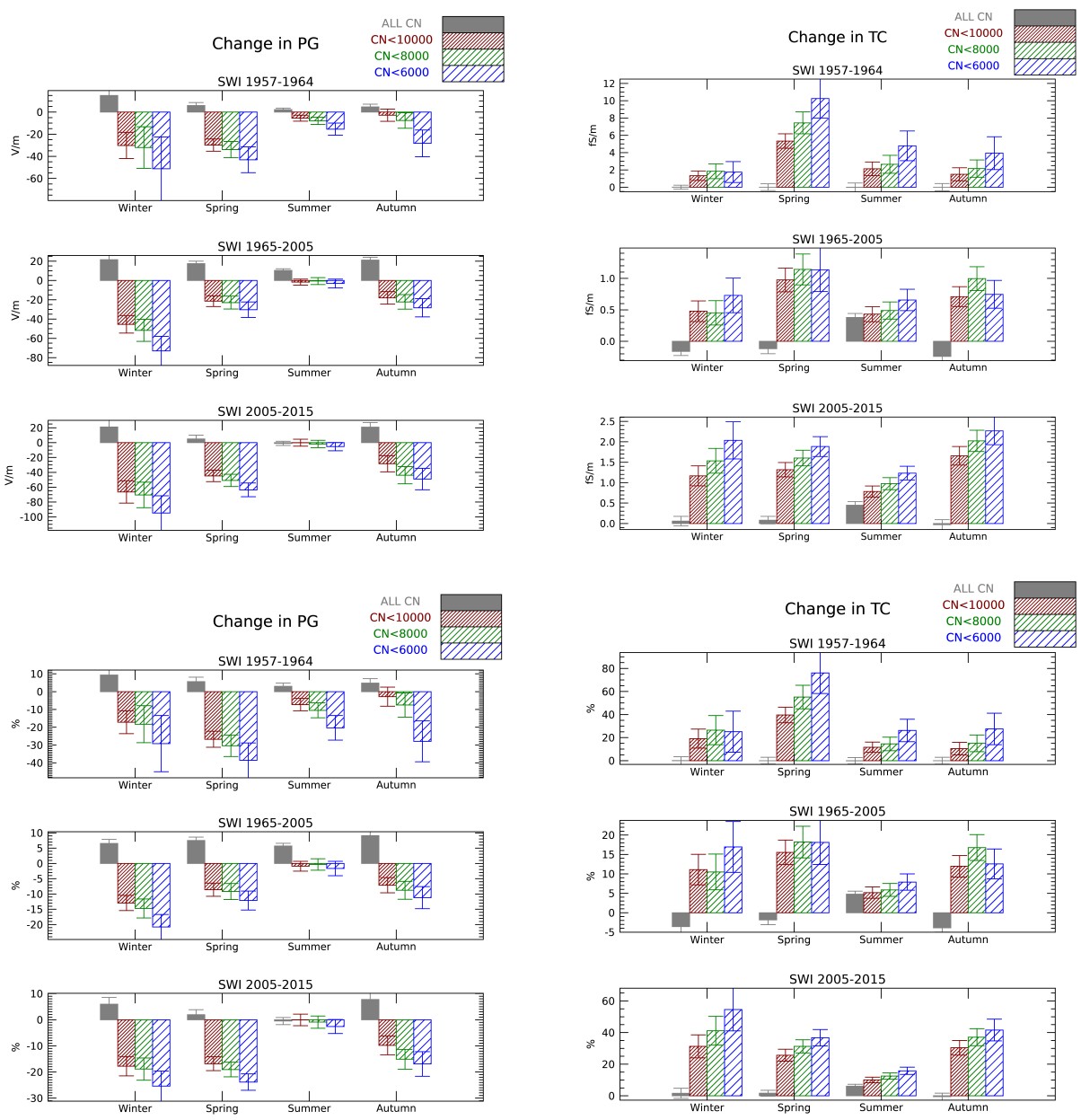

**Figure 13.** Seasonal variations of change in fair wethaer the potential gradient, PG, and total conductivity, TC, due to limited condensation nuclei concentration CN in the three considered periods (in 1965-2005 total conductivity equals double positive conductivity). Error bars indicate one standard error. Upper panels - absolute change. Bottom panels - relative change.

POTENTIAL GRADIENT     TOTAL CONDUCTIVITY    CONDENSATION NUCLEI

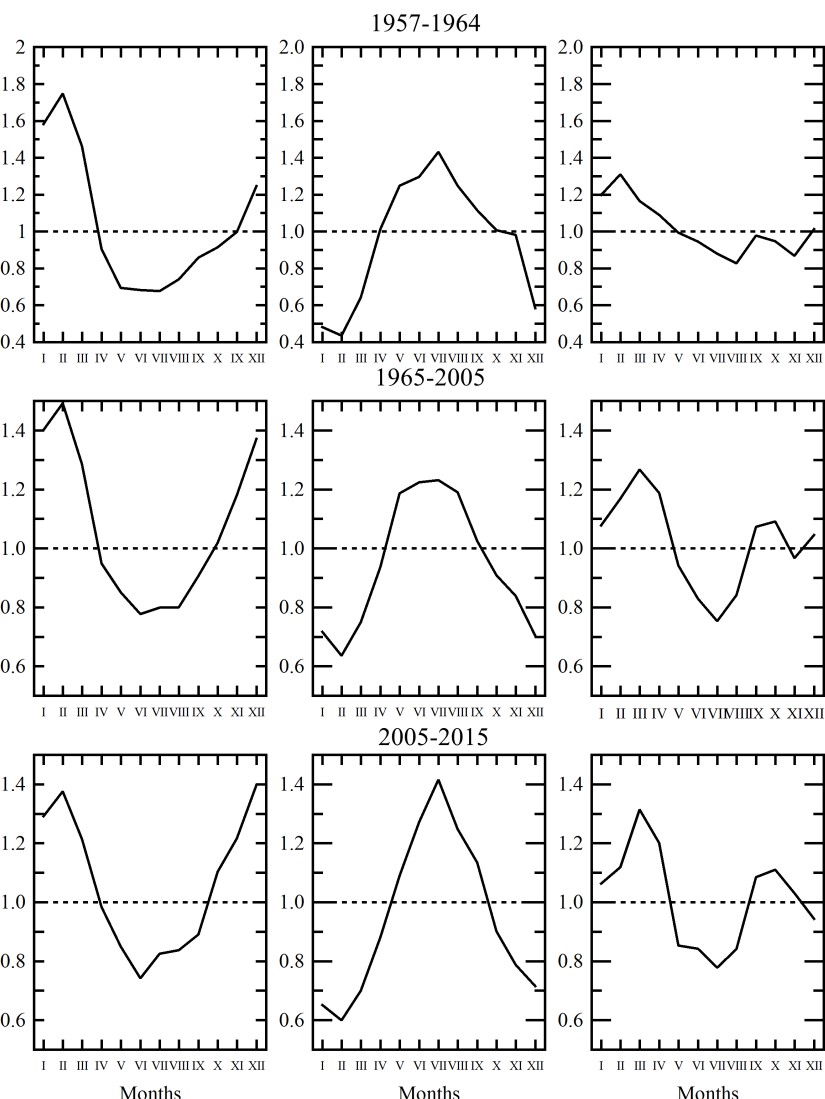

**Figure 14.** Average annual variation of fair weather potential gradient, PG, total air conductivity, TC and concentration of condensation nuclei, CN, in the three considered periods (in 1965-2005 total conductivity equals double positive conductivity).

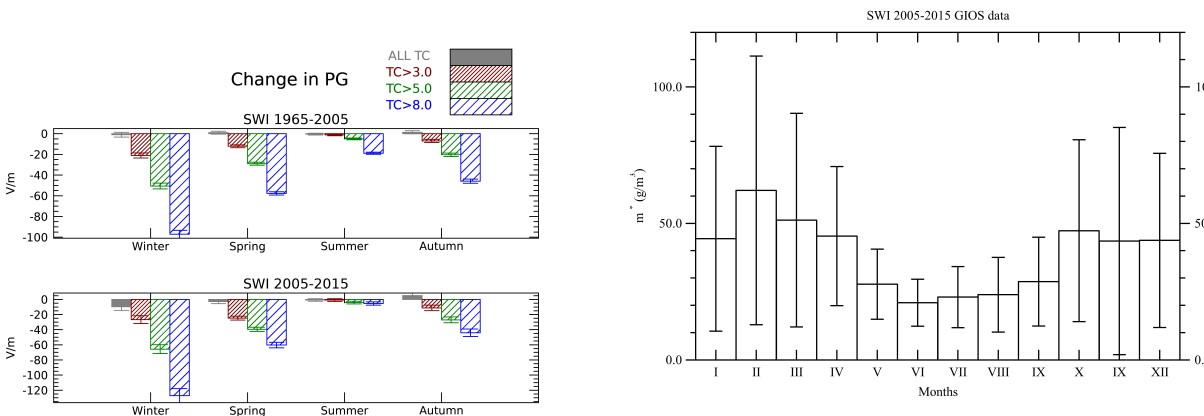

**Figure 15.** a) Seasonal variations of change in the potential gradient PG at three threshold values for the total air conductivity 3, 5 and 8 fS/m, respectively (in 1965-2005 total conductivity equals double positive conductivity). Error bars indicate one standard error. b) Monthly averages and standard deviations of PM10 concentration at Świder for the years 2005-2015 (GIOŚ data). Error bars indicate one standard deviation.