# Peer review of "Diurnal, seasonal and annual variations of fair weather atmospheric potential gradient, and effects of reduced number concentration of condensation nuclei on PG and air conductivity from long term atmospheric electricity measurements at Świder, Poland"

_Annales Geophysicae, 2024_

## Referee Comment (RC1)

**Review of " Analysis of diurnal, seasonal and annual variations of fair weather atmospheric potential gradient at reduced number concentration of condensation nuclei from long-term measurements at Swider, Poland" by I. Pawlak, A. Odzimek, D. Kepski and J. Tacza**

The main goal of this study, the assessment of the seasonal variation of the global electrical circuit from a polluted continental location, is very worthwhile and strongly appreciated, and from that standpoint every effort should be made to get the work published. The true seasonal variation of the DC global circuit is still not firmly established, even in measurements of the ionospheric potential (with contradictory seasonal variations from Muhleisen/Fishcer and from Markson). My main concern, already communicated to the second author, is the limited approach taken here: the measurement of condensation nuclei to characterize the atmospheric medium, rather than the measurement of air conductivity with a pair of Gerdien tubes. In the following, suggestions are made for improving the present approach, but in the end we suggest a conductivity approach that rests on observations readily available to the authors at the same Swider location (by Marek Kubicki).

**Summary: Consider for publication after major revisions (and possible inclusion of conductivity measurements)**

**Substantive issues:**

**(1) Characterization of the medium with a CN counter**

For reasons of time, the authors have been reluctant to get involved with the Swider conductivity measurements, and instead have chosen to rely on a CN counter. (The distinction between large and small ions of atmospheric electricity is not mentioned.) If the large ion population is reliably measured with the CN counter, then the air conductivity can be inferred (though this is not the best approach to obtaining air conductivity, as it is an indirect one). Unfortunately, the documentation on what is being measured with the Scholz counter is thin, even to the point of not disclosing what supersaturation value is achieved. It would also be valuable to know the instrument response to clean oceanic air but that is of course not easily obtained. In any case, a big improvement in the characterization of the Scholz counter is essential here.

**(2) The conductivity model used here**

Section 6 describes a conductivity model, but without sufficient details to thoroughly check its viability and origin. Equation (1) represents this model, but this is not an equation found in Tinsley and Zhou (2006). It may be an equation taken from Israel's text, but that is not identified. I for one do not recognize equation (1) from available references, though the inverse relationship between conductivity and N is reasonable. In addition, all parameters used here should be properly quantified and justified. One piece of evidence that this conductivity model is not working properly (even if equation (1) is taken at face value) is a simple check on Ohm's Law and air-earth current. One need only check equation (2) numerically (though it should be born in mind that the GEC air-earth current may vary annually). For winter, a value $\sigma$ = 2.28 x 10^-15 and E = 370 V/m, J = 0.84 pA, too small by at least a factor of two. For summer, $\sigma$ = 1.76 x 10^-15 and E = 370 V/m, J = 0.65 pA, and so too small by a factor greater than three. The evidence here is that the conductivity model is giving too small a conductivity, and that

inference is backed up by the large values of N coming out of the CN counter (with values per cc larger than ones typically reported in the literature, even for cities, see Chalmers (1967). The authors should make these points and these calculations. They also need to take a careful look at their conductivity model.

**(3) The arbitrary CN threshold of 10,000 per cc**

This threshold in CN is mentioned repeatedly (lines 8, 45, 99, 103, 111, in captions for Figures 7 and 8, lines 177, 341 and 347), but it is not made clear why this value was selected. No reference is given for justification. The suggestion is that the authors are seeking a characterization of the medium in cleaner conditions, but again the best parameter for that purpose is the Gerdien-measured conductivity, since this quantity is dominated by small ions with mobilities orders of magnitude larger than those for large ions/CN). In the end, the selection of this threshold is not resolving the main troublement at present (lines 13-14 of the Abstract and lines 360-362 of the Conclusions).

**(4) The main troublement of the paper**

The authors' main troublement, linked directly to the important interest in the annual variation in the source of the global electrical circuit, is that the seasonal variation in N (and conductivity inferred from the model that is N-dependent) is quite small in comparison with the potential gradient, leaving the impression that NH winter is still dominating the DC global circuit, as Lord Kelvin had inferred (probably incorrectly) more than a century ago. Based on the weight of the evidence now available, something is wrong with using the CN measurements to infer the true air conductivity.

**(5) The suggested resolution of the troublement**

The good news here is that long-term Gerdien tube measurements of the air conductivity are also available at Swider and have been presented in earlier unrefereed work by Kubicki et al. (2007), in a work that is cited in the present manuscript (page 2) but not elaborated on in the present context. The air conductivity is always dominated by the small ions (whose mobilities are orders of magnitude larger because of their small size, but these are not the ions measured with typical CN counters). Figure 1b of Kubicki et al. shows that the wintertime conductivity of air is reduced by a factor of two in comparison with summer, and is inverse with the measured seasonal change in potential gradient, making the seasonal variation in air-earth current much less than either conductivity or PG. The most important point here is that the seasonal change in air conductivity is MUCH LARGER than the variation of CN, the main source of the authors' troublement.

One last aspect of Kubicki et al. (2016) that also has relevance is Figure 1a, showing that the seasonal variation of dust exceeds a factor of 5, and so shows the largest seasonal variation of all. It is conceivable that the dust (however it is measured, and this is not explained in this brief abstract) is dominating the removal of small ions over the seasonal cycle. One must also be aware however (from Figure 6 of the reviewed work) that the true seasonal variation of CN is not captured by the 3-hour per day sampling of CN, a serious shortcoming.

**Summary:**

When air conductivity measurements at Swider are considered, and the authors can surely do that in a revised submission, the main troublement of the manuscript is removed. (This finding would not conflict

with the Kubicki et al findings because the seasonal issue was not focused on in that work.)  The revised findings would then support the general conclusions of Adlerman and Williams (1996) that the wintertime maximum in potential gradient is caused by the enhanced pollution expected in wintertime at extratropical locations such as Poland and the UK of Lord Kelvin.

End review

Earle Williams

April 1, 2024

---

## Author Comment (AC1)

Response to Reviewer #1 - attachment

1.  Modification to the conductivity model

Correction has been made to ion recombination coefficient now set to $\alpha$ = 1.3E-6 cm3/s, and ion production set to 2.5 $s^{-1}$ $cm^{-3}$. We want to further simplify the presentation of the model by showing the results of the ion conductivity change in just one table, when the relative proportions of water soluble and soot components vary (a small, constant contribution of insoluble aerosol of 500 $cm^{-3}$ is also assumed). We set the threshold values of soot and water soluble aerosol concentrations at 26000 $cm^{-3}$ and 6000 $cm^{-3}$, respectively. The first value corresponds to highest average aerosol concentrations in the winter (Fig. 4, Table 2), and the second value refers to the lowest concentrations measured at Świder (~4000 occur too, but they are very rare). Average winter situation may correspond to the ratio of the 10-20% vs 90-80% of water soluble and soot components which give the total CN concentration of 22500-24500 $cm^{-3}$ , since the average observed is ~20900 $cm^{-3}$, and summer conditions to the ratio of 60-70% vs 40-30%, i.e. 14500-12500 $cm^{-3}$ , since the average observed is ~15300 $cm^{-3}$ is (Table 2, Fig. 4). In the fifth column we give the effective attachment rate $\beta_{eff}$ = ($\beta$N)/N which varies in the range from 1.07 to 3.24 x $10^{-6}$ $cm^3$ $s^{-1}$. Sapkota and Varshneya (1990) mention that Hoppel predicts a range of 0.8 to 3.0 of $\beta_{eff}$ for the continental aerosol, so these are reasonable values. In the last column we give twice the experimental average value of polar (positive) conductivity calculated from newly digitised 1965-205 data: 4.4 ±0.2 x $10^{-15}$ S/m 4.4 ±0.2 for winter and 8.0±0.2 x $10^{-15}$ S/m for summer. While we adjusted q to give the right winter conductivity ~4.32-4.43 x $10^{-15}$ S/m, the model summer conductivity 4.97-5.13 x $10^{-15}$ S/m remains too low compared with the observations. It is likely that there are other constituents of the aerosol and dust that cause a larger difference between the winter and summer conductivities. The conductivity value very much depends on both $\beta_{eff}$ N and q, with $\beta_{eff}$ depending on the distribution of the sizes of the aerosol particles. In particula, the insoluble component also plays an important part through the high attachment rate. These may also vary between the summer and winter, and in the model they are constant. More analysis, and observational data from Świder are needed to develop a more realistic conductivity model, particularly of the aerosol size distributions.

| Water-Sol uble [%] | Soot [%] | Total concentration [$cm^{-3}$] | $\beta$N [$10^{-2}$ $s^{-1}$] | $\beta_{eff}$ [$10^{-6}$ $cm^3$ $s^{-1}$] | Sigma [$10^{-15}$ S/m] | Average observed conditions and conductivity value* [$10^{-15}$ S/m] |
|---|---|---|---|---|---|---|
| 0% | 100% | 26500 | 2.85 | 1.07 | 4.20 | |
| 10% | 90% | 24000 | 2.77 | 1.13 | 4.32 | |
| 20% | 80% | 22500 | 2.70 | 1.20 | 4.43 | WINTER 4.4 ±0.2 |

| 30% | 70% | 20500 | 2.62 | 1.28 | 4.56 | |
|---|---|---|---|---|---|---|
| 40% | 6 0% | 18500 | 2.55 | 1.38 | 4.69 | |
| 50% | 50% | 16500 | 2.48 | 1.50 | 4.82 | |
| 60% | 40% | 14500 | 2.40 | 1.66 | 4.97 | |
| 70% | 30% | 12500 | 2.33 | 1.86 | 5.13 | SUMMER 8.0±0.2 |
| 80% | 20% | 10500 | 2.26 | 2.15 | 5.29 | |
| 90% | 10% | 8500 | 2.18 | 2.57 | 5.47 | |
| 100% | 0% | 6500 | 2.11 | 3.24 | 5.66 | |

\* The value given equals twice the seasonal average of the positive (polar) conductivity calculated from hourly values reported in the observatory yearbooks (in fair weather conditions).

2. Annual variation of the positive conductivity at Świder.

We can now include the plot of the annual variation of positive conductivity observed at Świder.

---

## Author Response (AR1)

We thank the Reviewers for the insightful and helpful comments and for the careful reading of the manuscript.

Response to the comments made by Earle Williams, the Reviewer #1.

*The main goal of this study, the assessment of the seasonal variation of the global electrical circuit from a polluted continental location, is very worthwhile and strongly appreciated, and from that standpoint every effort should be made to get the work published. The true seasonal variation of the DC global circuit is still not firmly established, even in measurements of the ionospheric potential (with contradictory seasonal variations from Muhleisen/Fishcer and from Markson). My main concern, already communicated to the second author, is the limited approach taken here: the measurement of condensation nuclei to characterize the atmospheric medium, rather than the measurement of air conductivity with a pair of Gerdien tubes. In the following, suggestions are made for improving the present approach, but in the end we suggest a conductivity approach that rests on observations readily available to the authors at the same Swider location (by Marek Kubicki).*

*Summary: Consider for publication after major revisions (and possible inclusion of conductivity measurements)*

Reply: We thank the Reviewer for the careful and critical reading of the manuscript and the constructive comments made. After completing the digitisation of all Świder atmospheric electric 1965-2005 data, we were also able to present new results that include analysis of the annual variation of the conductivity measured at Świder (by the Gerdien tube). More details and a new plot are given in an added section of the manucript.

Response to substantive issues:

*(1) Characterization of the medium with a CN counter*

*For reasons of time, the authors have been reluctant to get involved with the Swider conductivity measurements, and instead have chosen to rely on a CN counter. (The distinction between large and small ions of atmospheric electricity is not mentioned.) If the large ion population is reliably measured with the CN counter, then the air conductivity can be inferred (though this is not the best approach to obtaining air conductivity, as it is an indirect one). Unfortunately, the documentation on what is being measured with the Scholz counter is thin, even to the point of not disclosing what supersaturation value is achieved. It would also be valuable to know the instrument response to clean oceanic air but that is of course not easily obtained. In any case, a big improvement in the characterization of the Scholz counter is essential here.*

Reply: In this study we wanted to concentrate rather on PG and aerosol number concentration as the PG is the most commonly observed atmospheric electric parameter worldwide, and there are more and more observations of aerosol concentrations, so other stations could make a similar analysis to investigate the results at their location.

We admit the information about the instrumentation for aerosol concentration measurements is limited as it is very scarce in the observatory yearbooks. We would like to add more details about the measuring apparatus which we found in the literature, in the archive materials of the Institute and from the observatory staff. A Scholz counter is a type of condensation counter constructed by Scholz as an improvement of the Aitken counter (McMurry 2000), designed to measure the concentration of condensation and nuclei nearly the total concentration of aerosol. At Świder, the main part of the small Scholz counter was a brass cylindrical chamber with a volume of 102 cm3 and a height of 4 cm,

with which the adiabatic expansion ratio of 1:1.25 could be achieved. The Scholz counter allows measurements in a wide range of CN concentrations from 5 to 960 000 particles in $cm^3$. We have not found any information on the experimental error, however, it should not be higher than the experimental error of the Aitken counter which is about 10%. The Scholz counter was used by the end of 1982 and after which a photoelectric counter was used as a more convenient (automatic) replacement. It was based on the construction of a Verzár counter which had the condensation chamber of 680 $cm^3$ in the volume.

*(2) The conductivity model used here*

*Section 6 describes a conductivity model, but without sufficient details to thoroughly check its viability and origin. Equation (1) represents this model, but this is not an equation found in Tinsley and Zhou (2006). It may be an equation taken from Israel's text, but that is not identified. I for one do not recognize equation (1) from available references, though the inverse relationship between conductivity and N is reasonable. In addition, all parameters used here should be properly quantified and justified. One piece of evidence that this conductivity model is not working properly (even if equation (1) is taken at face value) is a simple check on Ohm's Law and air-earth current. One need only check equation (2) numerically (though it should be born in mind that the GEC air-earth current may vary annually). For winter, a value σ = 2.28 x 10^-15 and E = 370 V/m, J = 0.84 pA, too small by at least a factor of two. For summer, σ = 1.76 x 10^-15 and E = 370 V/m, J = 0.65 pA, and so too small by a factor greater than three. The evidence here is that the conductivity model is giving too small a conductivity, and that inference is backed up by the large values of N coming out of the CN counter (with values per cc larger than ones typically reported in the literature, even for cities, see Chalmers (1967). The authors should make these points and these calculations. They also need to take a careful look at their conductivity model.*

Reply: We apologise for giving the wrong reference to Eq. 1. Tinsley and Zhou (2006) in their Eq. 3 give a differential equation, the solution of which is our Eq. 1. This equation can also be found in Schonland (1932), Makino and Ogawa (1985), among others. The input parameters for the conductivity model are also taken from other published models of the air electrical conductivity (which use the same Eq. 1), such as Makino and Ogawa (1985), Sapkota and Varshneya (1990), Tinsley and Zhou (2006), Kulkarni (2022). Every parameter used is based on empirical models (e.g. ion production) or other experimental data. The ion mobility of 1.5 $cm^2$ $s^{-1}$ $V^{-1}$ seems a realistic value for both positive and negative atmospheric ions at altitudes of up to 15 km according to Swider (1988). The ion recombination coefficient of 1.4 x $10^{-6}$ $cm^3$ $s^{-1}$ was used by Kulkarni (2022), similarly to Makino and Ogawa (1985). Ion production by cosmic rays of the order of 1.0 $s^{-1}cm^{-3}$ is appropriate for the production at the ground level, and of the order of 1.0 $s^{-1}cm^{-3}$ for the production by radioactivity from radon (8.6 $s^{-1}$ $cm^{-3}$ in Makino and Ogawa (1985) over land). In the correction we would like to update the ion production value to 2.2 $s^{-1}$ $cm^{-3}$ which gives more realistic values of the conductivity at Świder. We also need to correct ion recombination coefficient α since our calculations were made with α = 1.3 x $10^{-7}$ $cm^3$ $s^{-1}$. We leave the values of the beta coefficient of ion attachment rate calculated following the method of Tinsley and Zhou (2006) which uses the aerosol parameters of Hess et al. (1998). Hess et al. (1998) use constant model distributions of the aerosol constituents (soot, water soluble, insoluble and others) while in reality these distributions may also vary depending on seasons and day. In general, it is still a simple conductivity model which still needs to rely on several input parameters for various processes present such as the ion production, the rates of ion recombination and ion attachment to aerosol particles. For these parameters we use representative values used in other conductivity models while in reality these values may vary over a wider range, and in fact to have realistic model of local conductivity we need more observations to

derive the required parameters. We modified further the model as compared to our first reply.

*(3) The arbitrary CN threshold of 10,000 per cc*

*This threshold in CN is mentioned repeatedly (lines 8, 45, 99, 103, 111, in captions for Figures 7 and 8, lines 177, 341 and 347), but it is not made clear why this value was selected. No reference is given for justification. The suggestion is that the authors are seeking a characterization of the medium in cleaner conditions, but again the best parameter for that purpose is the Gerdien-measured conductivity, since this quantity is dominated by small ions with mobilities orders of magnitude larger than those for large ions/CN). In the end, the selection of this threshold is not resolving the main troublement at present (lines 13-14 of the Abstract and lines 360-362 of the Conclusions).*

Reply: We have given additional references (Landsberg 1938, Schonland 1953, Mohnen and Hidy 2010).

*(4) The main troublement of the paper*

*The authors' main troublement, linked directly to the important interest in the annual variation in the source of the global electrical circuit, is that the seasonal variation in N (and conductivity inferred from the model that is N-dependent) is quite small in comparison with the potential gradient, leaving the impression that NH winter is still dominating the DC global circuit, as Lord Kelvin had inferred (probably incorrectly) more than a century ago. Based on the weight of the evidence now available, something is wrong with using the CN measurements to infer the true air conductivity.*

Reply: Yes, the interest in the annual variation of the global electrical circuit was for us an important issue and one for the purpose of which we carried out the investigation, however, in the first place we wanted to investigate whether, having a long-term series of PG and CN measurements we would be able select the conditions of low aerosol concentration, and whether this could reveal a different character of the annual variation of the PG. It did not, however, since the PG still had a maximum over NH winter months, and we never said that corresponded to the annual maximum of the GEC. Perhaps we have not emphasised this conclusion of our paper enough. We agree that using only 3 hours from each day may not represent total seasonal or diurnal variation but it does represent some characteristic points that capture the variation of CN during a day (Fig. 6).

*(5) The suggested resolution of the troublement*

*The good news here is that long-term Gerdien tube measurements of the air conductivity are also available at Swider and have been presented in earlier unrefereed work by Kubicki et al. (2007), in a work that is cited in the present manuscript (page 2) but not elaborated on in the present context. The air conductivity is always dominated by the small ions (whose mobilities are orders of magnitude larger because of their small size, but these are not the ions measured with typical CN counters). Figure 1b of Kubicki et al. shows that the wintertime conductivity of air is reduced by a factor of two in comparison with summer, and is inverse with the measured seasonal change in potential gradient, making the seasonal variation in air-earth current much less than either conductivity or PG. The most important point here is that the seasonal change in air conductivity is MUCH LARGER than the variation of CN, the main source of the authors' troublement.*

*One last aspect of Kubicki et al. (2016) that also has relevance is Figure 1a, showing that the seasonal variation of dust exceeds a factor of 5, and so shows the largest seasonal variation of all. It is*

*conceivable that the dust (however it is measured, and this is not explained in this brief abstract) is dominating the removal of small ions over the seasonal cycle. One must also be aware however (from Figure 6 of the reviewed work) that the true seasonal variation of CN is not captured by the 3-hour per day sampling of CN, a serious shortcoming.*

Reply: The conductivity measurements have been carried at Świder, however the observatory yearbooks report only the value of the positive conductivity. By now we have digitised the whole 1956-2005 series and are able to present some analysis of the data. The annual variation of the conductivity confirms earlier results of Kubicki et al. with a minimum in NH winter months and twice higher values in the summer. There is also a distinct effect of conductivity to lowering the affecting CN concentrations but there is no very clear seasonal effect.

 *Summary:  When air conductivity measurements at Swider are considered, and the authors can surely do that in a revised submission, the main troublement of the manuscript is removed. (This finding would not conflict with the Kubicki et al findings because the seasonal issue was not focused on in that work.) The revised findings would then support the general conclusions of Adlerman and Williams (1996) that the wintertime maximum in potential gradient is caused by the enhanced pollution expected in wintertime at extratropical locations such as Poland and the UK of Lord Kelvin.*

Reply: Thank you for the detailed consideration given to our manuscript. We added results including an analysis of the conductivity for the purpose of investigating its annual variation and conclusions for the GEC annual variation. We concluded the conditions of low condensation nuclei conditions does not change the character of the annual variation of the PG at Świder.

*Response to Reviewer #2*

*The paper contains a reinterpretation of historic data to investigate the relationship between aerosol concentration and atmospheric potential gradient. Many investigations similar to this have used proxies or estimates for aerosol concentrations, so the inclusion of real aerosol data makes this a useful addition to the literature.*

*In my view, the manuscript could aid the reader more my adding more technical information on the measurements and some additional statistical analysis. The site description may be included in the references, but it would benefit the manuscript to put pertinent information on the measurement site and places of interest nearby. For example, how near is the measurement site to major population centers, roads, the coast or industry. This would aid in understanding potential sources of aerosol.*

Reply: Indeed, we could add additional information about the measurement site. The PAS Geophysical Observatory in Świder is located in the central part of Poland in the Warsaw suburban area, about 25 km south-east. Świder used to be a popular holiday and health resort village located on the Świder river. The distance to the nearest urban centre, which is the district town of Otwock, is 2.5 km. There is no major industry in the area but there are local anthropogenic sources of air pollution from heating, very typical for these suburban conditions. The architecture is dominated by residential buildings and mainly includes single and multifamily houses. The Observatory is located is less populated area nearer the river and covers about 7 ha. It includes the main office and three observation pavilions, two residential buildings are in a distance. The entire area is surrounded by trees (predominantly pine trees), with several clearings. In one of these clearings of an area of

approximately 1 ha, one pavilion and the station's instruments for atmospheric electricity and meteorology observations are located.

*The measurement instrumentation is listed but more detail on the instruments used would also be helpful. Of particular importance is the measurement range, sensitivity and error of the aerosol measurement. I believe more detail of the aerosol instrumentation needs to be repeated within the manuscript, as the references are not easily available. Please could you include details of what kind of aerosol counter is used, the size range that are counted and details of inlet tubing.*

Reply: We admit the information on the details of the apparatus is scarce as it is also very scarce in the observatory yearbooks, especially in regard to the technical details of the counters. We would like to add more details about the measuring apparatus which we found in the literature, in the archive materials of the Institute and from the observatory staff. We also add some details of the methodology and schedule of observations.

Measurements of concentration of condensation nuclei (CN) in Geophysical Observatory in Świder between 1965 and 2005 were carried out using two counters: a small Scholz counter and a photoelectric CN counter built in the observatory and using chamber of a Verzár counter as a base. The measurement method used in both counters was the process of condensation of water vapor on atmospheric aerosol particles present in the measurement chamber, followed by a quantitative analysis of the resulting mist droplets. We intend to add literature references where these counters are described in more detail, e.g. McMurry (2000). Observations were performed three times a day between 5:50 GMT and 6:20 GMT (till 1971), 6:10 GMT and 6:30 GMT (afterwards), then between 11:00 GMT and 11:30 GMT, and between 19:00 GMT and 19:30 GMT (till 1971), 18:10 UT – 18:30 UT afterwards. These three periods of observations are referred to in the yearbooks as 06, 12, and 18 GMT or UT, respectively.

At Świder, the main part of the small Scholz counter was a brass cylindrical chamber with a volume of 102 cm3 and a height of 4 cm, and the adiabatic expansion ratio of 1:1.25 could be achieved. Measurements were performed within the clearing of the meteorological station using the suction method, an air sample with a volume of 1 cm3 was taken at a height of 1 m above the ground surface. Measurements with a Scholz counter should be repeated a few times, and one measurement takes several minutes. The Scholz counter allows measurements in a wide range of CN concentrations from 5 to 960 000 particles in cm3. We have not found any information on the experimental error, however, it should not be higher than the experimental error of the Aitken counter which is about 10%.

Since January 1983, the measurements have been performed with the photoelectric CN counter that was placed inside a measurement pavilion. The air samples were collected from the outside of the building at a height of 1 m above the ground. The suction of air was made through a 1 m long rubber pipe using an electric rotational pump. According to the yearbooks the counter enabled measurement of the concentration of CN whose radius ranged from 0.005 to 10 μm. The measuring range of the counter was 4500 to 850 000 CN in 1 cm3 of the air. The basic measurement of the number of CN took place in a cylindrical chamber filled with the tested air sample of the volume equal to 680 cm3. Estimates of the number of droplets were obtained using a photoelectric counter system by measuring the extinction of light. The electronic circuit system was built (also patented) by Stanisław Warzecha. The measurement accuracy was 15%.

*In section 2.3, who assessed the criteria for fair weather, was this done by observatory staff, or more recently?*

The criteria for fair weather were assessed by the observatory staff on an ongoing basis. The information is also included in the yearbooks.

*Finally, I would suggest that the paper could be strengthened by adding some more statistical testing to the analysis. For example, it is claimed that limiting the condensation nuclei concentrations does not significantly change the distributions of corresponding potential gradient values – was a test used to prove the difference is not significant? When comparing the shape of distributions, could you consider using a non-parametric distribution test, eg, Kolmogorov-Smirnov test?*

Reply: Thank you for pointing this out. We have tried to support our conclusions by using a non-parametric Kolmogorov-Smirnov (KS) and U Mann-Whitney (MW) tests. They have been used to calculate the statistical significance of differences between the PG distributions considering the decreasing number of CN. In regard to the distributions shown in Fig. 2 and in Fig. 7, both KS and MW tests indicated statistically significant differences (at the significance level of 0.05) between the PG-CN all and PG-CN<10000 distributions for the whole year as well as in the spring, autumn and winter. In case of the variations shown in Fig. 11, the KS test indicated statistically significant differences between the PG populations for all CNs and CN<10000, in January and February only. The MW test indicated such in April and November as well. When comparing PG and CN all with PG and CN<8000 the results of both tests indicate the statistically significant difference also in December, similarly to PG and CN<6000.

*A few small corrections:*

*Section 2.1, there was a change if instrumentation, was any cross checking done to verify if there was a change in response?*

Reply: We don't know exactly if any cross-checking has been done. In the yearbooks there is no information about it.

*In Figure 1, is there any exclusion criteria before a data point can be included, for example, is there a minimum number of fair weather data points required for the month to be included?*

Reply: No criterion regarding the minimum number of points for the months has been applied. Our intention was to use all the possible hourly values obtained during fair weather conditions.

*In Figure 2 (and some subsequent figures) the values separated by commas are not clear which plot they belong too and this should be made explicit.*

Reply: We added more description in the figures captions.

*In Figure 6 can you clarify how long the measurement is that the means at 6h, 12h and 18h are?*

Reply: As we mentioned, a single measurement lasted several minutes (about 5 min), and for any observation usually several single measurements were carried out and the final result was the average value calculated from these several measurements.

*Line 47: in the statement 'less than 10,000' could you say what size these particle fall under?*

Reply: The above statement refers to the number of CN below 10000 cm-3, we can suggest that there are particles from the whole measuring range (from 0.005 μm to 10 μm).

*Line 228: What is the justification for the values given for the variable in equation 1, are they based on measured quantities?*

Reply: Our intention was to use representative values of the necessary parameters, several of which are required for even such a simple conductivity model. Parameters of such order are used by various conductivity models which use values based on observational data or empirical models. These are the models of Makino and Ogawa (JGR 1985), Sapkota and Varshneya (1990), Tinsley and Zhou (2006), or Kulkarni (2022). Ion production by cosmic rays of the order of $1.0\ s^{-1}cm^{-3}$ is appropriate for the production at the ground level, and of the order of $1.0\ s^{-1}cm^{-3}$ for the production by radioactivity of radon. In the correction we would like to update the ion production value to $2.2\ s^{-1}cm^{-3}$ which gives more realistic values of the conductivity. The ion mobility of $1.5\ cm^2\ s^{-1}\ V^{-1}$ seems a realistic value for both positive and negative atmospheric ions at altitudes of up to 15 km according to Swider (1988).

*Line 230: What is the justification in assuming both mobilities are equal, as they are often not different to each other. Is it possible to run a sensitivity analysis to see what a difference would make?*

Reply: As mentioned above the same value of $(1.5 +- 0.3)\ cm^2\ s^{-1}\ V^{-1}$ could be used for both positive and negative ions. A change of the mobility by $0.1\ cm^2\ s^{-1}$ causes a change in the conductivity by about $5 \times 10^{-17}$ S/m (5%) which is rather small in comparison with the effect of other parameters considered.

*Line 242: Please provide more information on where and when these size distributions were measured.*

The information is taken from Kubicki et al. (2016), and the distributions were measured at the Świder Observatory after 2010. The data have not been published, however.

*Line 254: Why was relative humidity excluded?*

Reply: The effect of humidity would complicate the conductivity model even more, and we are unable at the moment to use a practical conductivity model that takes it into account. The aerosol concentration distribution parameters of Hess et al. (1998) used for the calculation of the ion attachment rate are given for the relative humidity of 50%. We consider it is appropriate for average fair weather conditions, even though observational values may cover a wider range.

*Some smaller changes:*

*Line 133, change "except of" to "except in"*

*Line 176, superscript should be -3*

*Line 273: Should this read 1.3 times by?*

Reply: Thank you for the detailed consideration given to our manuscript. We have corrected the mistakes.

---

## Referee Report (RR1)

**Second review of "Analysis of diurnal, seasonal and annual variations of fair weather atmospheric potential gradient at reduced number concentration of condensation nuclei from long-term measurements at Swider, Poland",** by Izabela Pawlak, Anna Odzimek, Daniel Kepski and Jose Tacza

On the occasion of the Workshop on the Global Electrical Circuit in Warsaw, Poland earlier this month, I had good opportunity to meet with the coauthors of this manuscript to learn from them and to share my thoughts.  I am attaching (after the main review) here a copy of the comments I had put together on first looking at the revision.  That revision is much improved, reinforcing my earlier view that the paper is much improved and eventually deserves to be published.  But for the moment, I have some very specific suggestions for a path to that publication.

**Summary:  Consider for publication after major revision**

Suggestions for addressing remaining loose ends:

(1)  Need for bipolar conductivity

A big step forward was the accessing of the Swider positive conductivity.  But the very best quantity for treating the seasonal issue and the highly polluted Swider boundary layer in winter, one really wants the bipolar (total) conductivity, and based on my separate discussion with Marek Kubicki, these bipolar data are available in the Swider archive.  The authors should get them and include them and come up with the best explanation for the seasonal variation of the electric field, and then make a firm judgment about local versus global influence/manifestation.  In the present version, the conclusions are not firm.

(2)  The expansion ratio threshold for the CN counter

Wilson (1897) showed (in work that eventually won him the Nobel Prize in physics) that when the expansion ratio of 1.25 was exceeded, a rainlike condensation on small ions appeared in his cloud chamber.  This is essentially the threshold the authors have provided in supplying valuable new information about their CN counter.  But they need to try to establish whether  the counter was designed with the CTR Wilson finding in mind, so that they were just short of activating on small ions.  I am not quite sure how that can be accomplished, but some effort should be made.  At the very least, the Wilson (1897) work and threshold should be mentioned in the revised text.

(3)  The CN threshold of 10,000 per cc

Through the response to initial reviews, and through discussion in Warsaw, the authors have conveyed to me their general strategy in their selection of this CN threshold:  an effort to achieve conditions sufficiently clean so as to achieve a globally representative measurement at Swider.  (In their response to the reviews, they cited Landsberg (1938), Schonland (1953) and Mohnen and Hidy (2010), and now I have had a chance to consult the latter two references.  Perhaps the best information in this context is contained in Table 2 of Mohnen and Hidy, and I encourage them to continue to focus on this information.

Having considered all of this information, my general reaction is that the value of 10,000 per cc is still quite large given the perceived objective of the authors.  To be more specific, 10,000 per cc is large than the average values for 7 of 9 categories listed in Table 2.  So the authors can say that they have selected a threshold value less than the means for "Town" and "City" in Table 2, but they need to say that this

general level is still "polluted" and that the finding that they are still unable to make globally representative measurements of the global electrical circuit from Swider is very believable. Remember the initial success of the Carnegie Institution ocean measurements of electric field (and CN!) in achieving global representativeness, but their values for CN were of the order of 100 per cc, and so two orders of magnitude less than the authors' selected threshold. More discussion in the manuscript is needed about the full range of CN conditions one can have.

(4) Reference to Adlerman and Williams (1996)

The authors have been interested in checking on the role of aerosol variation on the seasonal variation of the global circuit. It would be valuable, after they checked carefully the seasonal variations of the Gerdien bipolar conductivity at Swider, to ascertain whether the inferences made by Adlerman and Williams (1996) are valid.

(5) Globally representative measurements at Swider

This issue came up at the recent Workshop on the Global Electrical Circuit and is deserving of some additional discussion in the present context. Two key points are worthy of discussion. First, Marek Kubicki informed me about days in which the Gerdien bipolar conductivity is essentially constant over 24 hours. This evidence for a fixed medium (yes, in polluted conditions and low values of conductivity) may provide an advantage to seeing global signals, and deserves additional attention. (The more common behavior at Swider is for highly diurnally variable conductivity.) Second, the boundary layer over land is often stabilized at nighttime by virtue of temperature inversions. This situation, well known to meteorologists but not exploited by atmospheric electricians in earlier work, might also provide a condition when convective transport of electric space charge is zero, and for which Ohm's law is fulfilled. This makes it more likely that a measurement in this special time interval would be globally representative, because local perturbations are strongly suppressed.

I am now wandering beyond the scope of the manuscript, but the authors may benefit from giving these ideas further thought.

End review

Earle Williams

September 22, 2024

**Comments communicated to the authors prior to the Workshop:**

September 6, 2024

Below are my thoughts in blue text, in response to your responses on my review.

Brief responses to Response to Reviewer comments document

(Sorry I have that document only as PDF so will indicate here where I am inserting my remarks to respond. I will send return that document to you when I send these comments.)

Paragraph after "Summary: Consider for publication after…

Bravo for bringing the Swider conductivity to the table. This is a BIG improvement in the manuscript.

(1)  Characterization of the medium with a CN counter

First paragraph of Reply:

I understand about interest in concentrating on the PG and aerosol, but if you want to address important scientific issues, then the emphasis on conductivity is also important.  The interpretative attention on conductivity could be extended, in my opinion.

Second paragraph:  This is a big improvement in the documentation of the instrumentation, though one key item about the expansion ratio is still needed.  (Reviewer #2 also emphasized the need for greater elaboration on the instrumentation.)

With regard to "the adiabatic expansion ratio of 1: 1.25 could be achieved"

An accurate estimate here is quite critical, based on the earlier findings of C.T.R. Wilson with his cloud chamber.  See in particular Wilson (1897), page 286, which states: "For adiabatic expansion to result in condensation in saturated air free of all foreign nuclei, we have seen that the final volume must exceed 1.252 times the initial volume."  As shown later by Wilson, and in his Nobel Prize work, that amounted to a 4-fold supersaturation, which is just sufficient for nucleation on negative ions.  So it is important to know for sure whether the expansion ratio in the reported work was consistently below this critical threshold established by Wilson.  Maybe your instrument was designed that way, but this should be stated.

(2)  The conductivity model used here

The discussion here is very much improved.  Thanks very much for your efforts.

(3)  The arbitrary CN threshold of 10,000 per cc

You aren't responding in any detail here to my comment, other than giving references I have not yet consulted.  (The revised text has more info, to be sure.)   My take-away here is that you are trying to find sufficiently clean air so that the measurements of PG at Swider can become globally representative.  To justify such a procedure, you need to take care to compare this selected threshold with values one expects in places where globally representative measurements of PG can sometimes be achieved, most notably over clean oceans and at Vostok and Concordia stations in Antarctica.  Recent analysis of many years of the Vostok PG data (paper being submitted for review now) do support a NH summer maximum of the GEC, contrary to the PG maximum in wintertime that you are documenting at Swider.

(4)  The main troublement of the paper

After your Reply, first sentence:

Yes, but if the interest in the annual variation in the global electrical circuit is an important issue, why isn't this aspect given more attention in the revision?  (It got more attention in the first version, but now you are (appropriately) bringing in the conductivity.)  The polluted atmosphere as Swider is aliasing the GEC, and makes the PG measurements there unrepresentative of global variations.

"It did not, however, since the PG still has a maximum over winter months…"

OK, but you are not giving sufficient attention to the role of the polluted atmosphere at Swider in causing that maximum, in my opinion. (For me, Swider is the classic example of how wintertime pollution prevents access to globally representative measurements.)

"Perhaps we have not emphasized this conclusion of our paper enough."

I completely agree with this assessment, and think some changes are in order so that it is appropriately emphasized.

> (5) The suggested resolution of the troublement

After Reply: The conducitivity measurements have been carried at Swider…"

Did Marek use previously only positive values, in the works that were discussed? (I will find out soon enough as I am meeting with Marek on Monday in Warsaw.) I had thought that Marek had access to bipolar conductivity from the Gerdien, an instrument I remember seeing at Swider on occasion of my last visit there in 2015.

"There is also a distinct effect of conductivity to lowering the affecting CN concentrations but there is no clear seasonal effect"

Isn't this tantamount to saying that even when you select data for which the CN are below this pre-selected threshold of 10,000 per cc, that you cannot undo the effect of the continental pollution at Swider in destroying a global signature? The attempt to find sufficiently clean air at Swider to enable global representativeness is not possible because those times are few and far between.

"We conclude the conditions of low condensation nuclei conditions does not change the character of the annual variation of the PG at Swider."

This finding needs to be better expressed in the revised manuscript, in my opinion, though I confess to not having read carefully the entire revision yet. You need to underline the lack of global representativeness in my opinion.

I am sending these comments with an aim to set up a discussion when I see you both next week in Warsaw, but only if you are interested, of course.

Regards
Earle

---

## Referee Report (RR2)

**Third Review of "Diurnal, seasonal and annual variations of fair weather atmospheric potential gradient, and effects of reduced number concentration of condensation nuclei on PG and air conductivity from long term atmospheric electricity measurements at Swider, Poland " by I. Pawlak, A. Odzimek, Daniel Kepski, and José Tacza**

Moving this paper forward to publication has proven to be challenging. I have exchanged with the authors (Pawlak and Odzimek) and shared my reviews with them (and visited with them in Poland at a Workshop on the Global Circuit and at the recent AGU meeting) to assist with a difficult topic. The encouragement to interact more closely with Marek Kubicki has been useful in getting the involvement with the conductivity variable (PC), but has also led us into this "dust" topic (see below) which becomes a new complication for Swider, though one that could also be addressed with CN analysis, since dust particles are also CN. Further exchange with Marek is needed here.

The author's observation location at Swider is a polluted continental site but you have tools to investigate that. The authors have long had an idealistic goal of getting globally representative measurements of the GEC by compensation in making observations in conditions of reduced CN (<10,000 per cc), and so conditions much closer to clean maritime ones. Such conditions are unfortunately infrequent. From the time of my first review, my recommendation has been to shift attention from CN to Gerdien conductivity, as the latter is a quantity more closely connected with GEC behavior and the values remain valid even in highly polluted conditions. Here I will summarize again the difficulties with bringing in the CN in this study, and which have still not been overcome.

- (1) The authors are not yet addressing the important "dust" issue with the CN measurements. Yes the appeal to the earlier Kubicki et al. (ICAE, 2007) work raises dust, so to speak. Otherwise, bringing in Kubicki et al more strongly is helpful here, because that study makes use of the all-important Gerdien conductivity data (which are rarely available in atmospheric electrical observations), and conductivity is much more important for the global circuit interest than the CN observations. The authors are now working with that archived quantity. This is valuable even if it is only unipolar conductivity data. Marek Kubicki has additional info on the dust (D) quantity that should be followed up.
- (2) The use of reduced-CN data, intended to enable study of the PG observations in cleaner conditions, is still not reaching appropriately clean conditions. 10,000 per cc is not clean. The authors recognize this difficulty in multiple places (one example is lines 61-65).
- (3) The authors lack a reliable means to estimate the electrical conductivity with the measured CN observations. Please correct me if my claim is incorrect. This problem stands in the way of drawing firm conclusions in this study. The absence of a simple relationship between conductivity (well measured with the Gerdien tube) and CN (measured with the CN counter) is clear from Figure 13 in the revised manuscript. The connection between conductivity and CN needs to be more quantitative than what is expressed in lines 481-482. The authors quote changes in PG (of order tens of %) as CN values are decreased from 10,000 per cc, but they do not use their conductivity model to predict what these changes should be. Even rough agreement could be used to declare partial success with the conductivity model.

Another key interest in this work is the seasonal variation in the DC global electrical circuit. I mentioned in an earlier review that this variation was not well-established, largely because of contamination from local effects. Since that time, recent work by Russian scientists (Slyunyaev et al. 2024 in JGR) has

demonstrated a northern hemisphere summer maximum by making use of Vostok, Antarctica measurements of potential gradient which are not contaminated by aerosol/CN and or dust, and so the results are convincing. I have been a reviewer of this work. These findings also raise the bar in verifying the seasonal variation of the GEC at Swider, in polluted conditions.

Summary: The authors should be encouraged to produce a revised manuscript that gives greater attention to the conductivity observations than the CN observations, and which sheds further light on the physical role of the "dust" at Swider. See further details below.

Additional comments on the revised manuscript appear below.

Lines 13-15 This is the dust issue and raises a key question that is left unanswered by the revised manuscript, even when CN is returned to as a topic of key interest. Dust particles should also serve as CN, so why does the CN counter not see the large seasonal variation evidenced in the work of Kubicki et al. (2007)?

Line 37 This question on the seasonal variation of the GEC has now been investigated in considerable detail by the Russians and papers in JGR should be appearing soon. The NH summer maximum in the DC GEC is supported by Vostok measurements of PG, running for many years.

Line 45 The statement about the air conductivity is unclear.

Line 50 It is challenging to find "low levels of nuclei number" at Swider as we have discussed. This situation thwarts the authors' main interest in finding conditions needed for a look at global representativeness. The improvement here is that the authors are now facing up to what conditions are needed.

Lines 64-65 I agree, and this thwarts the main goal of the study.

Line 101 You should say that the air conductivity is dominated by small ions, but all ions contribute. The air conductivity should also be influenced by the presence of dust, which is also aersosol. This aspect should be investigated further for Swider.

Section 2.3 Important new documentation of CN measuring equipment has now been added, including maximum supersaturation attained.

Line 142 Normally one is using "foul" for bad weather conditions.

Line 166 10,000 per cc is still a polluted condition.

Lines 171-172 Why is summer more polluted than winter? What is the seasonal variation of the dust?

Appeal to Kubicki et al. (2007) is needed here.

Line 482 The authors do not answer this question about whether PG data could ever be used to infer the annual variation of the GEC. The likely reason is that one never has a sufficiently clean condition to have globally representative results.

Line 485 Other aerosol types: the authors should strive to address the nature of "dust" in the earlier study by Kubicki et al. (2007), who first addressed the seasonal variations in Gerdien conductivity. Why

isn't this "dust" measured with CN counters? Kubicki shows an annual variation of dust substantially larger than CN at Swider. Why? Further interpretation is needed here.

My best recommendation, and in keeping with my initial review: Make use of the high-quality Gerdien tube data the authors have now demonstrated access to and go beyond what Kubicki et al (2007) achieved with the seasonal variations. The authors have made progress with organizing the Swider Gerdien data but they need to improve on the interpretation of the seasonal behavior. You will have better temporal resolution than you had with the CN data and you will be investigating a quantity (conductivity) more closely connected with the DC GEC than the CN observations. This effort may also expose more information about the dust component of aerosol (emphasized by Marek Kubicki in 2007) and its quantitative impact on the conductivity. How was that dust quantity measured? It is not explained in the abstract. One wants to understand why the seasonal change in conductivity is much larger than the variation of CN.

Adlerman and Williams (1996) is discussed in the Introduction, but after looking at the seasonal variations in conductivity and PG, the authors do not return to the seasonal aerosol variation is a plausible explanation for the seasonal variation in PG. And since the authors now have seasonal variation in both PG and PC, why don't they have a look at the air-earth current to see if this is compatible with a NH summer maximum in storm source currents?

If the authors wish to emphasize the CN observations in this study, they need to tie them in more closely with conductivity than is achieved at present.

End review Earle Williams Febuary 6, 2025

---

## Author Response (AR2)

**Responses to Reviewers**

**We thank both Reviewers for the careful reading of the manuscript and helpful comments.**

**Response to the comments made by Earle Williams, the Reviewer #1, in the second review:**

On the occasion of the Workshop on the Global Electrical Circuit in Warsaw, Poland earlier this month, I had good opportunity to meet with the coauthors of this manuscript to learn from them and to share my thoughts. I am attaching (after the main review) here a copy of the comments I had put together on first looking at the revision. That revision is much improved, reinforcing my earlier view that the paper is much improved and eventually deserves to be published. But for the moment, I have some very specific suggestions for a path to that publication.

Suggestions for addressing remaining loose ends:

**(1) Need for bipolar conductivity**

A big step forward was the accessing of the Swider positive conductivity. But the very best quantity for treating the seasonal issue and the highly polluted Swider boundary layer in winter, one really wants the bipolar (total) conductivity, and based on my separate discussion with Marek Kubicki, these bipolar data are available in the Swider archive. The authors should get them and include them and come up with the best explanation for the seasonal variation of the electric field, and then make a firm judgment about local versus global influence/manifestation. In the present version, the conclusions are not firm.

Response: According to your suggestion, in addition to the conductivity data from 1965-2005, we also used bipolar (total) conductivity for the periods: 1957-1964 and 2005-2015. The annual variation of the total conductivity (TC) is shown in Figure 13.

**(2) The expansion ratio threshold for the CN counter**

Wilson (1897) showed (in work that eventually won him the Nobel Prize in physics) that when the expansion ratio of 1.25 was exceeded, a rainlike condensation on small ions appeared in his cloud chamber. This is essentially the threshold the authors have provided in supplying valuable new information about their CN counter. But they need to try to establish whether the counter was designed with the CTR Wilson finding in mind, so that they were just short of activating on small ions. I am not quite sure how that can be accomplished, but some effort should be made. At the very least, the Wilson (1897) work and threshold should be mentioned in the revised text.

Response: After consultations we can state that the counter was designed to be as close as possible to CTR Wilson. According to your comment Wilson's publication and the threshold value were mentioned in the revised text.

**(3) The CN threshold of 10,000 per cc**

Through the response to initial reviews, and through discussion in Warsaw, the authors have conveyed to me their general strategy in their selection of this CN threshold: an effort to achieve conditions sufficiently clean so as to achieve a globally representative measurement at Swider. (In their response to the reviews, they cited Landsberg (1938), Schonland (1953) and Mohnen and Hidy (2010), and now I have had a chance to consult the latter two references. Perhaps the best information in this context is contained in Table 2 of Mohnen and Hidy, and I encourage them to continue to focus on this information.

Having considered all of this information, my general reaction is that the value of 10,000 per cc is still quite large given the perceived objective of the authors. To be more specific, 10,000 per cc is large than the average values for 7 of 9 categories listed in Table 2. So the authors can say that they have selected a threshold value less than the means for "Town" and "City" in Table 2, but they need to say that this general level is still "polluted" and that the finding that they are still unable to make globally representative measurements of the global electrical circuit from Swider is very believable. Remember the initial success of the Carnegie Institution ocean measurements of electric field (and CN!) in achieving global representativeness, but their values for CN were of the order of 100 per cc, and so two orders of magnitude less than the authors' selected threshold. More discussion in the manuscript is needed about the full range of CN conditions one can have.

Response: You are right that the threshold value of 10,000 per cc is a value that still reflects polluted conditions, however taking into account the Świder location (near the Warsaw) and possible emission, the CN concentration about 10,000 per cc are very probable. It is worth mention that it is the upper threshold of CN concentration analyzed in the article. To show "cleaner" conditions we used also threshold of 8,000 and 6,000, and 4,000 per cc (see Table 1). However, as we reduced the CN concentration the number of PG values decreases to some extent.

**(4) Reference to Adlerman and Williams (1996)**

The authors have been interested in checking on the role of aerosol variation on the seasonal variation of the global circuit. It would be valuable, after they checked carefully the seasonal variations of the Gerdien bipolar conductivity at Swider, to ascertain whether the inferences made by Adlerman and Williams (1996) are valid.

**Response: We have additionally digitsed dat from period 1957-2005 and reworked data from 2005-2015 to analyse the bipolar conductivity changes.**

**(5) Globally representative measurements at Swider**

This issue came up at the recent Workshop on the Global Electrical Circuit and is deserving of some additional discussion in the present context. Two key points are worthy of discussion. First, Marek Kubicki informed me about days in which the Gerdien bipolar conductivity is essentially constant over 24 hours. This evidence for a fixed medium (yes, in polluted conditions and low values of conductivity) may provide an advantage to seeing global signals, and deserves additional attention. (The more common behavior at Swider is for highly diurnally variable conductivity.) Second, the boundary layer over land is often stabilized at nighttime by virtue of temperature inversions. This situation, well known to meteorologists but not exploited by atmospheric electricians in earlier work, might also provide a condition when convective transport of electric space charge is zero, and for which Ohm's law is fulfilled. This makes it more likely that a measurement in this special time interval would be globally representative, because local perturbations are strongly suppressed.

Response: we agree but resolving the issue needs at least continuous measurements of CN, dust concentrations and particle size distributions. Perhaps some other type of analysis with selection of special conditions would be more useful.

We would like to thank you for the valuable discussions during the workshop, for all comments and suggestions regarding the manuscript.

In addition to the problems indicated above, we have corrected some presented numbers after we made a small correction to our dataset. We have also tried to improve spelling and grammar.

---

## Author Response (AR3)

**Responses to Reviewers**

Manuscript: "Diurnal, seasonal and annual variations of fair weather atmospheric potential gradient, and effects of reduced number concentration of condensation nuclei on PG and air conductivity from long term atmospheric electricity measurements at Swider, Poland" by I. Pawlak, A. Odzimek, D. Kepski, and J.Tacza

**We would like to thank all Reviewers of this manuscript for very careful reading and valuable comments.**

Response to the comments made by Earle Williams, the Reviewer #1, in the third review:

*Moving this paper forward to publication has proven to be challenging. I have exchanged with the authors (Pawlak and Odzimek) and shared my reviews with them (and visited with them in Poland at a Workshop on the Global Circuit and at the recent AGU meeting) to assist with a difficult topic. The encouragement to interact more closely with Marek Kubicki has been useful in getting the involvement with the conductivity variable (PC), but has also led us into this "dust" topic (see below) which becomes a new complication for Swider, though one that could also be addressed with CN analysis, since dust particles are also CN. Further exchange with Marek is needed here.*

*The author's observation location at Swider is a polluted continental site but you have tools to investigate that. The authors have long had an idealistic goal of getting globally representative measurements of the GEC by compensation in making observations in conditions of reduced CN (<10,000 per cc), and so conditions much closer to clean maritime ones. Such conditions are unfortunately infrequent. From the time of my first review, my recommendation has been to shift attention from CN to Gerdien conductivity, as the latter is a quantity more closely connected with GEC behavior and the values remain valid even in highly polluted conditions. Here I will summarize again the difficulties with bringing in the CN in this study, and which have still not been overcome.*

Response:

We sincerely thank the reviewer for the detailed comment and the long-standing engagement with our work. We would like to clarify a key point that may have led to a misunderstanding. Our intention in this study was not to obtain globally representative measurements of the Global Electric Circuit (GEC). Rather, our specific objective was to investigate the role of condensation nuclei (CN) in modulating potential gradient (PG) variations at a polluted continental site like Świder.

By examining PG under varying CN concentrations, including instances when CN values were relatively low as possible at the site (generally <10,000 cm$^{-3}$). We aimed to understand the extent to which local aerosol loading influences PG at this location. This was not motivated by wanting to generalize our results to globally representative GEC conditions, but instead to characterize the local impact of aerosols on PG behaviour, and check in what way this may alter the annual variation.

 *(1) The authors are not yet addressing the important "dust" issue with the CN measurements. Yes the appeal to the earlier Kubicki et al. (ICAE, 2007) work raises dust, so to speak. Otherwise, bringing in Kubicki et al more strongly is helpful here, because that study makes use of the all-important Gerdien conductivity data (which are rarely available in atmospheric electrical observations), and conductivity is much more important for the global circuit interest than the CN observations. The authors are now*

*working with that archived quantity. This is valuable even if it is only unipolar conductivity data. Marek Kubicki has additional info on the dust (D) quantity that should be followed up.*

Response:

Indeed, the dust data for the Świder observatory were collected during at least some of the periods analyzed in our work. However, the old data are not digitized, and the more recent data are the result of cooperation with the State Sanitary Inspectorate, are not publicly available and their inclusion in the work would require a significant expansion of work and staff, if possible at all. Introducing such significant changes and analyzing another large data set at this stage of the publication process is impossible. Additionally, most of the information on dust until 2010 is available only on a daily basis - filters were changed once a day, obtaining the weight of dust deposited on the filter. The result of these observations would be difficult to compare with the measurements of condensation nuclei performed 3 times a day and presented in our work, especially knowing that relationship between condensation nuclei number and particulate mass is complex (Leng et al. 2014).

Leng, C., et al. "Variations of cloud condensation nuclei (CCN) and aerosol activity during fog–haze episode: a case study from Shanghai." Atmospheric Chemistry and Physics 14.22 (2014): 12499-12512. https://acp.copernicus.org/articles/14/12499/2014/acp-14-12499-2014.pdf

At present we cite some old results published by Haberko (1961) which provides some justification for the study.

*(2) The use of reduced-CN data, intended to enable study of the PG observations in cleaner conditions, is still not reaching appropriately clean conditions. 10,000 per cc is not clean. The authors recognize this difficulty in multiple places (one example is lines 61-65).*

Response:

We appreciate the reviewer's observation regarding the limitations of using CN < 10,000 cm$^{-3}$ as a threshold for cleaner conditions. We fully acknowledge that this level does not correspond to truly clean air in a global or maritime context. However, this threshold was selected pragmatically, based on the data availability at the Świder station. Actual range of CN considered is between 5000-8000 cm$^{-3}$, which still is polluted compared to Arctic or Antarctic.

As noted in lines 255-259, reducing the dataset further by applying a stricter CN threshold would result in a significant number of data gaps. This would compromise the statistical robustness and temporal continuity necessary for meaningful analysis of PG variations. Therefore, the chosen threshold represents a balance between aiming for relatively lower aerosol conditions and maintaining sufficient data coverage for the study's objectives.

*(3) The authors lack a reliable means to estimate the electrical conductivity with the measured CN observations. Please correct me if my claim is incorrect. This problem stands in the way of drawing firm conclusions in this study. The absence of a simple relationship between conductivity (well measured with the Gerdien tube) and CN (measured with the CN counter) is clear from Figure 13 in the revised manuscript. The connection between conductivity and CN needs to be more quantitative than what is expressed in lines 481-482. The authors quote changes in PG (of order tens of %) as CN values are decreased from 10,000 per cc, but they do not use their conductivity model to predict what*

*these changes should be. Even rough agreement could be used to declare partial success with the conductivity model.*

*Response:*

*We think are simplified model was developing in the right direction, and we hope to make progress on it.*

*Another key interest in this work is the seasonal variation in the DC global electrical circuit. I mentioned in an earlier review that this variation was not well-established, largely because of contamination from local effects. Since that time, recent work by Russian scientists (Slyunyaev et al. 2024 in JGR) has demonstrated a northern hemisphere summer maximum by making use of Vostok, Antarctica measurements of potential gradient which are not contaminated by aerosol/CN and or dust, and so the results are convincing. I have been a reviewer of this work. These findings also raise the bar in verifying the seasonal variation of the GEC at Swider, in polluted conditions.*

Response:

We thank the reviewer for this valuable comment and for highlighting the recent findings by Slyunyaev et al. (2024), which indeed provide an important contribution to our understanding of the seasonal variation in the global electric circuit (GEC) under roughly clean conditions.

However, we would like to clarify that the main objective of our study was not to establish or verify the 'real' seasonal variation of the DC GEC itself. Rather, our aim was to investigate how seasonal changes in aerosol loading - quantified via CN concentrations - affect the seasonal behavior of the PG at a polluted continental site such as Świder. In this context, we explored whether reducing the CN concentration (e.g., to values <10,000 cm$^{-3}$) could help reveal clearer patterns in PG variability that may be obscured by local pollution effects.

 We agree that measurements at cleaner sites, such as Vostok, are better suited for identifying unambiguous seasonal signals in the global circuit. Nevertheless, we believe that studies like ours offer complementary insights by quantifying the extent to which local aerosol variability modulates PG measurements, which is essential for interpreting long-term records from continental sites.

Moreover, even in Antarctic there may be some seasonal change in the concentration of aerosol which may have an effect on the annual PG variation. We hope this was taken into account.

*Summary: The authors should be encouraged to produce a revised manuscript that gives greater attention to the conductivity observations than the CN observations, and which sheds further light on* the physical role of the "dust" at Swider. *See further details below.*

Response:

We thank the Reviewer for the suggestion. After more digitization efforts we have considered the conductivity and effects of dust, and what is even more clear from the preliminary analysis, the situation is complicated, and there must be other processes like the convection affecting PG at a site like this. Without more comprehensive study and measurements, it could be very difficult to infer any GEC variation from the PG variation at such a site.

*Additional comments on the revised manuscript appear below.*

*Lines 13-15 This is the dust issue and raises a key question that is left unanswered by the revised manuscript, even when CN is returned to as a topic of key interest. Dust particles should also serve as CN, so why does the CN counter not see the large seasonal variation evidenced in the work of Kubicki et al. (2007)?*

Response:

Theoretically, in the case of atmospheric electricity, the most important charge carriers are single molecules and this is why we focused on condensation nuclei measurements. It is known that seasonal fluctuations in the number of aerosol particles are not large in Central Europe (Asmi et al. 2011), but the mass of particulate matter is much higher in winter, which results in frequent occurrence of smog (Czernecki et al. 2017). The high number of aerosol in summer (which balances the CN amount from pollution in winter) can be explained by secondary aerosol formation with the participation of volatile organic compounds and high insolation events (Dall'Osto et al., 2018). Obviously, it is true that "dust" also serve as CN. It may be very interesting to study which particle sizes and origin influence the most electric field strength, but unfortunately we are not able to perform such an analysis for the study period. We conclude that there may be a dust fraction invisible to our CN counters.

Asmi, Ari, et al. "Number size distributions and seasonality of submicron particles in Europe 2008–2009." Atmospheric Chemistry and Physics 11.11 (2011): 5505-5538. https://acp.copernicus.org/articles/11/5505/2011/acp-11-5505-2011.pdf

Czernecki, Bartosz, et al. "Influence of the atmospheric conditions on PM 10 concentrations in Poznań, Poland." Journal of Atmospheric Chemistry 74 (2017): 115-139. https://link.springer.com/content/pdf/10.1007/s10874-016-9345-5.pdf

Dall'Osto, Manuel, et al. "Novel insights on new particle formation derived from a pan-european observing system." Scientific reports 8.1 (2018): 1482. https://pmc.ncbi.nlm.nih.gov/articles/PMC5784154/pdf/41598_2017_Article_17343.pdf

*Line 37 This question on the seasonal variation of the GEC has now been investigated in considerable detail by the Russians and papers in JGR should be appearing soon. The NH summer maximum in the DC GEC is supported by Vostok measurements of PG, running for many years.*

Response:

Thank you for pointing us to the latest research results, although relating the results from such a different and clean environment is difficult to relate directly to the results obtained here.

*Line 45 The statement about the air conductivity is unclear.*

Response:

We clarified the sentence. Now it states: "while the air conductivity variability is higher in the summer" (line 47).

*Line 50 It is challenging to find "low levels of nuclei number" at Swider as we have discussed. This situation thwarts the authors' main interest in finding conditions needed for a look at global representativeness. The improvement here is that the authors are now facing up to what conditions are needed.*

Response:

We accept this remark and are aware that it is a certain limitation as to the usefulness of the Świder data for determining global changes in the electric field. We hope that now, in the revised version of the manuscript, this shortcoming has been clearly commented on.

*Lines 64-65 I agree, and this thwarts the main goal of the study.*

Response:

Unfortunately we are limited by the site characteristics. Nevertheless, we believe that this data matters and the conclusions from this work can be valuable to the community.

*Line 101 You should say that the air conductivity is dominated by small ions, but all ions contribute. The air conductivity should also be influenced by the presence of dust, which is also aersosol. This aspect should be investigated further for Swider.*

Response:

We admit that the emphasis of small ions here is not entirely correct. We have changed 'small ions' to 'ion mobility' in the text. We are aware of the predominance of small ions and that all ions affect conductivity, but we decided not to expand on this topic here.

In Section 2.3 important new documentation of CN measuring equipment has now been added, including maximum supersaturation attained.

*Line 142 Normally one is using "foul" for bad weather conditions.*

Response:

Thank you for pointing this out. We replaced "bad" by "foul".

*Line 166 10,000 per cc is still a polluted condition.*

*Response:*

*We are awared that 10000 is still not clean air conditions, but we decided for such threshold to separate days with exceptionally high particle counts from our analysis and see if this improves the relationship with the electric field.*

*Lines 171-172 Why is summer more polluted than winter? What is the seasonal variation of the dust?*

Response:

We believe that we explained it before. Probably this is the effect of secondary aerosol formation on sunny summer days. We think it is better to talk about higher concentration of aerosol particles than

pollution, because in winter the air is certainly more polluted. In summer, the greater part of the aerosol will be of natural origin, which we should not talk about as pollution.

*Appeal to Kubicki et al. (2007) is needed here.*

Response:

Although we are aware that in Kubicki et al. 2007 was mentioned secondary aerosol formation that may be referred also here, we decided not to cite it here as we just describe our results here. Reference to this effect and work of Kubicki was done in new line 455.

*Line 482 The authors do not answer this question about whether PG data could ever be used to infer the annual variation of the GEC. The likely reason is that one never has a sufficiently clean condition to have globally representative results.*

Response:

We want to emphasize again that this was not the main intention of the work. We do not agree with the second sentence. With almost 50 years of data, it is possible to find days with nearly perfect air quality to analyze global electric current, although it is true that Swider data as a whole for sure are not representative for annual potential gradient variability. Here we present seasonal variations between PG and CN trying to focus on relatively clean conditions.

*Line 485 Other aerosol types: the authors should strive to address the nature of "dust" in the earlier study by Kubicki et al. (2007), who first addressed the seasonal variations in Gerdien conductivity. Why isn't this "dust" measured with CN counters? Kubicki shows an annual variation of dust substantially larger than CN at Swider. Why? Further interpretation is needed here.*

Response:

Dust is also measured by CN counters. The average CN number is higher in winter than in summer at the Swider observatory, but the difference is not that large in number, but great in aerosol mass (which we know from data presented in new Figure 15). It seems that different aerosol types influence electric field in different ways. This may suggest that bigger (and heavier) particles present in winter have larger effect on electric field properties than we previously thought and we should study this effect more thoroughly in the future. There is also a possibility that we should consider not solely on CN as decisive factor, but take into account also e.g. air humidity.

*My best recommendation, and in keeping with my initial review: Make use of the high-quality Gerdien tube data the authors have now demonstrated access to and go beyond what Kubicki et al (2007) achieved with the seasonal variations. The authors have made progress with organizing the Swider Gerdien data but they need to improve on the interpretation of the seasonal behavior. You will have better temporal resolution than you had with the CN data and you will be investigating a quantity (conductivity) more closely connected with the DC GEC than the CN observations. This effort may also expose more information about the dust component of aerosol (emphasized by Marek Kubicki in 2007) and its quantitative impact on the conductivity. How was that dust quantity measured? It is not*

*explained in the abstract. One wants to understand why the seasonal change in conductivity is much larger than the variation of CN.*

Response:

We added new two Figures to analyse the effect of particulate matter (dust) on conductivity records and new text in section 7 to describe this new data. We hope that this will complement the previous results and provide more complete look at the Świder observatory data.

*Adlerman and Williams (1996) is discussed in the Introduction, but after looking at the seasonal variations in conductivity and PG, the authors do not return to the seasonal aerosol variation is a plausible explanation for the seasonal variation in PG. And since the authors now have seasonal variation in both PG and PC, why don't they have a look at the air-earth current to see if this is compatible with a NH summer maximum in storm source currents?*

Response:

In the revised version of the manuscript we paid more attention to the air conductivity data and their relationship with the variability in aerosol concentration and mass. We do not wish to extend this work further but we hope that the general relationship between aerosol conditions, air conductivity and potential gradient measurements is now more comprehensively explained in case of Świder, and the associated air pollution problem in this area. Analysis of conduction current density would extend this work enormously. In the revision we indicate that such an opportunity exists but it is also not free from problems.

*If the authors wish to emphasize the CN observations in this study, they need to tie them in more* closely with conductivity than is achieved at present.

*End review*

Response:

New figures and text was added to address this issue. In the two latest revisions the CN influence is analysed both in relation to the potential gradient and the air conductivity.

Thank you for all your valuable comments on the text. We believe that the corrections presented here thanks to your suggestions have made this work more valuable.